# Targeting cyclin-dependent kinases for the treatment of pulmonary arterial hypertension

Astrid Weiss[1,2,3,4,11], Moritz Christian Neubauer[1,2,3,4,11], Dinesh Yerabolu[1,2,3,4], Baktybek Kojonazarov[1,2,3,4], Beate Christiane Schlueter[1,2,3,4], Lavinia Neubert[4,5], Danny Jonigk[4,5], Nelli Baal[1,2,4,6], Clemens Ruppert[1,2,4], Peter Dorfmuller[4,7], Soni Savai Pullamsetti [2,3,4,8], Norbert Weissmann [1,2,3,4], Hossein-Ardeschir Ghofrani[1,2,3,4,9], Friedrich Grimminger[1,2,3,4,10], Werner Seeger [1,2,3,4,8,10] & Ralph Theo Schermuly [1,2,3,4]

Pulmonary arterial hypertension (PAH) is a devastating disease with poor prognosis and limited therapeutic options. We screened for pathways that may be responsible for the abnormal phenotype of pulmonary arterial smooth muscle cells (PASMCs), a major contributor of PAH pathobiology, and identified cyclin-dependent kinases (CDKs) as overactivated kinases in specimens derived from patients with idiopathic PAH. This increased CDK activity is confirmed at the level of mRNA and protein expression in human and experimental PAH, respectively. Specific CDK inhibition by dinaciclib and palbociclib decreases PASMC proliferation via cell cycle arrest and interference with the downstream CDK-Rb (retinoblastoma protein)-E2F signaling pathway. In two experimental models of PAH (i.e., monocrotaline and Su5416/hypoxia treated rats) palbociclib reverses the elevated right ventricular systolic pressure, reduces right heart hypertrophy, restores the cardiac index, and reduces pulmonary vascular remodeling. These results demonstrate that inhibition of CDKs by palbociclib may be a therapeutic strategy in PAH.

[1] Justus-Liebig-University Giessen (JLU), Aulweg 130, Giessen 35392, Germany. [2] Universities of Giessen and Marburg Lung Center (UGMLC), Giessen, Germany. [3] Cardio-Pulmonary Institute (CPI), EXC 2026, Project ID: 390649896, Giessen, Germany. [4] Member of the German Center for Lung Research (DZL), Giessen, Germany. [5] Institute of Pathology, Hannover Medical School (MHH), Carl-Neuberg-Strasse 1, Hannover 30625, Germany. [6] Institute for Clinical Immunology and Transfusion Medicine, University Hospital Giessen and Marburg (UKGM), Aulweg 128, Giessen 35392, Germany. [7] Department of Pathology, University Hospital of Giessen and Marburg (UKGM), Langhansstrasse 10, Giessen 35392, Germany. [8] Max Planck Institute (MPI) for Heart and Lung Research, Parkstrasse 1, Bad Nauheim 61231, Germany. [9] Department of Medicine, Imperial College London, London, UK. [10] University Hospital Giessen and Marburg (UKGM), Giessen, Germany. [11] These authors contributed equally: Astrid Weiss, Moritz Christian Neubauer. Correspondence and requests for materials should be addressed to R.T.S. (email: ralph.schermuly@innere.med.uni-giessen.de)

Pulmonary arterial hypertension (PAH) is a rapidly progressing disease of the lung vasculature with poor prognosis, ultimately leading to right heart failure and death. The remodeling of small pulmonary arteries represents an important pathological characteristic of PAH. The hyperproliferation and resistance to apoptosis exhibited by smooth muscle cells, endothelial cells, and fibroblasts result in pulmonary vascular obstruction, leading to increased pulmonary artery pressure and right ventricle overload[1]. The vascular lesions noted in patients with PAH consist primarily of pulmonary arterial smooth muscle cells, which exhibit a tumor-like phenotype similar to the hallmarks of cancer, where these cells exhibit uncontrolled replicative potential, self-sustaining growth signals, evasion of growth suppressors, resistance to apoptosis, and changes in cellular metabolism and inflammatory processes[2–6]. Currently available treatments focus primarily on vasodilation rather than on causal mechanisms of the disease and the structural remodeling processes.

Several members of the cyclin-dependent kinase (CDK) family orchestrate the complex events that drive the cell cycle. To ensure a coordinated transition through the cell cycle, activity of CDKs is controlled by their corresponding cyclin protein partners, the expression and activation of which are tightly regulated during cell cycle progression. Transition to the next phase depends on the activity of CDK1, CDK2, CDK4, and CDK6. Notably, transition from G1 to S phase is mediated by CDK4-cyclinD and CDK6-cyclinD, together with CDK2-cyclinE, all of which can phosphorylate and inactivate the retinoblastoma (Rb) protein, an anti-tumor and anti-proliferative protein that limits the transcription of E2F target genes, such as *CCNA2 (CyclinA2)* and *CDK1*. CDK1-cyclinA and CDK2-cyclinA drive cells through the S phase, whereas mitosis depends on CDK1-cyclinB[7,8]. The activity of these regulators is, in turn, regulated by endogenous CDK inhibitors that bind cyclin-CDK complexes, thereby causing G1 arrest, and thus limiting cell proliferation under unfavorable conditions[9]. The activity of the CDK–cyclin complex is frequently deregulated in cancer cells by a spectrum of mechanisms that include decreased levels of endogenous CDK inhibitors, leading to persistently elevated levels of active cyclin-CDK complexes, and thus uncontrolled cell growth[9,10].

Synthetic CDK inhibitors such as dinaciclib and palbociclib are currently investigated in clinical trials as potential anti-cancer drugs[9,11,12]. Dinaciclib is a selective and potent small-molecule inhibitor with clinical potential in both hematological and solid malignancies[13–15]. Dinaciclib inhibits CDK1, CDK2, CDK5, and CDK9 activity[8], and exhibits anti-proliferative effects within the low nanomolar range against most cancer cells[16,17]. Palbociclib has been recently approved for the treatment of advanced breast cancer, targeting CDK4 and CDK6[18]. We hypothesize that the hyperproliferation of abnormal pulmonary vascular cells can be blocked by CDK inhibitors. We report that the activity of several CDKs is upregulated in the pulmonary vasculature in PAH in human specimen and in two experimental models. Dinaciclib and palbociclib efficiently block cell cycle progression in isolated primary pulmonary smooth muscle and endothelial cells, which leads to a reduced structural remodeling process in vivo. Targeted inhibition of CDK4 and 6 by palbociclib in MCT, as well as Su5416/hypoxia treated rats results in an improvement of several pathophysiological parameters. Our findings support the further clinical development of these compounds for the treatment of PAH.

## Results

### CDK activity and expression in human PAH and animal models.
Primary human pulmonary arterial smooth muscle cells (HPASMCs) were isolated from the explanted lungs of healthy individuals or patients with idiopathic PAH (IPAH) that underwent lung transplantation. Cells were incubated for 48 h with basal media (BM) without any growth factors. To address the serine/threonine kinase activity in these cells, equal amounts of protein extracts were applied to specific peptide array chips (PamGene International BV, ´s-Hertogenbosch, The Netherlands). This methodology has been described as a screening tool that allows for the robust analysis of serine/threonine kinase activity from cells and tissues[19–22]. There was a clear difference in the mean pattern of peptide substrate phosphorylation comparing HPASMCs from healthy individuals and HPASMCs isolated from the lungs of IPAH patients (IPAH-HPASMCs) as illustrated in the heat map (Fig. 1a), indicating that under disease conditions, the activity of certain signaling pathways driven by distinct kinases is altered. In a two-group comparison between healthy HPASMCs and IPAH-HPASMCs, the de-regulation of phosphorylation for each peptide (Fig. 1b) represents the effect of changes in kinase activity. Based on those phosphorylation patterns, a computational upstream kinase analysis predicted increased activity of multiple kinases involved in cell cycle control in the IPAH-HPASMCs, namely CDK2, CDK4, CDK6, and CDK9 (Fig. 1c). The phosphorylation status of the individual peptide substrates of each of these kinases is illustrated (Fig. 1d). Although some heterogeneity in the samples was observed for each condition (Supplementary Fig. 1), a significant change in the pattern of phosphorylation was noted when comparing the healthy HPASMCs and IPAH-HPASMCs. Two commercially available inhibitors, dinaciclib (targeting CDK2 and CDK9) and palbociclib (targeting CDK4 and CDK6), were tested for anti-proliferative potential in (IPAH-)HPASMCs in vitro. To confirm downstream activation of the CDK-Rb-E2F signaling pathway, western blot (Fig. 1c) and real-time PCR analyses were undertaken for CDK expression (Supplementary Fig. 2) and subsequent transcription of two established target genes namely *CDK1* (Fig. 1f) and *CyclinA2 (CCNA2)* (Supplementary Fig. 2e). Here, upregulation of (P)-CDK2, CDK4, P-CDK6, and Cyclin D1 at the protein level, and an increase in *CDK1* and *CyclinA2* mRNA levels could be demonstrated, supporting increased activity of the CDK-induced Rb-E2F pathway in HPASMCs from IPAH patients. To confirm that the predicted increase in activity of CDK2, CDK4, CDK6, and CDK9 is due to an enhanced expression level under disease conditions, real-time PCR analyses were undertaken in isolated primary HPASMCs 48 h after starvation (Supplementary Fig. 2a–h) and in homogenates of explanted human lungs (Supplementary Fig. 2i–p). In HPASMCs (Supplementary Fig. 2a–d), as well as in lung homogenates from IPAH patients (Supplementary Fig. 2i–l), increased *CDK2* and *CDK6* mRNA levels were noted, whereas *CDK4* mRNA levels were only elevated in isolated cells, while *CDK9* mRNA levels remained unaffected. *CyclinA2 (CCNA2)* was the only CDK-regulating cyclin, which exhibited higher expression under disease conditions (Supplementary Fig. 2e, m). Similar findings concerning the expression of CDKs were noted in lung homogenates from experimental models of P(A)H: In the murine model of hypoxia-induced PH (3 weeks of hypoxia) (Supplementary Fig. 3a–d), only an increase in *CDK2* mRNA levels were observed (Supplementary Fig. 3a), whereas in the MCT rat model (5 weeks after MCT injection) (Supplementary Fig. 3e–h) and the Su/Hox rat model (Su5416-injection, 3 weeks hypoxia followed by 2 weeks re-exposure to normoxic conditions) (Supplementary Fig. 3i–l), a strong upregulation of expression of almost all CDKs was noted.

### The CDK-Rb-E2F-PCNA signaling cascade in human IPAH.
Immunohistological analyses were undertaken in serial sections of human lungs tissue for CDK2 (Fig. 2a), CDK4 (Fig. 2b), and

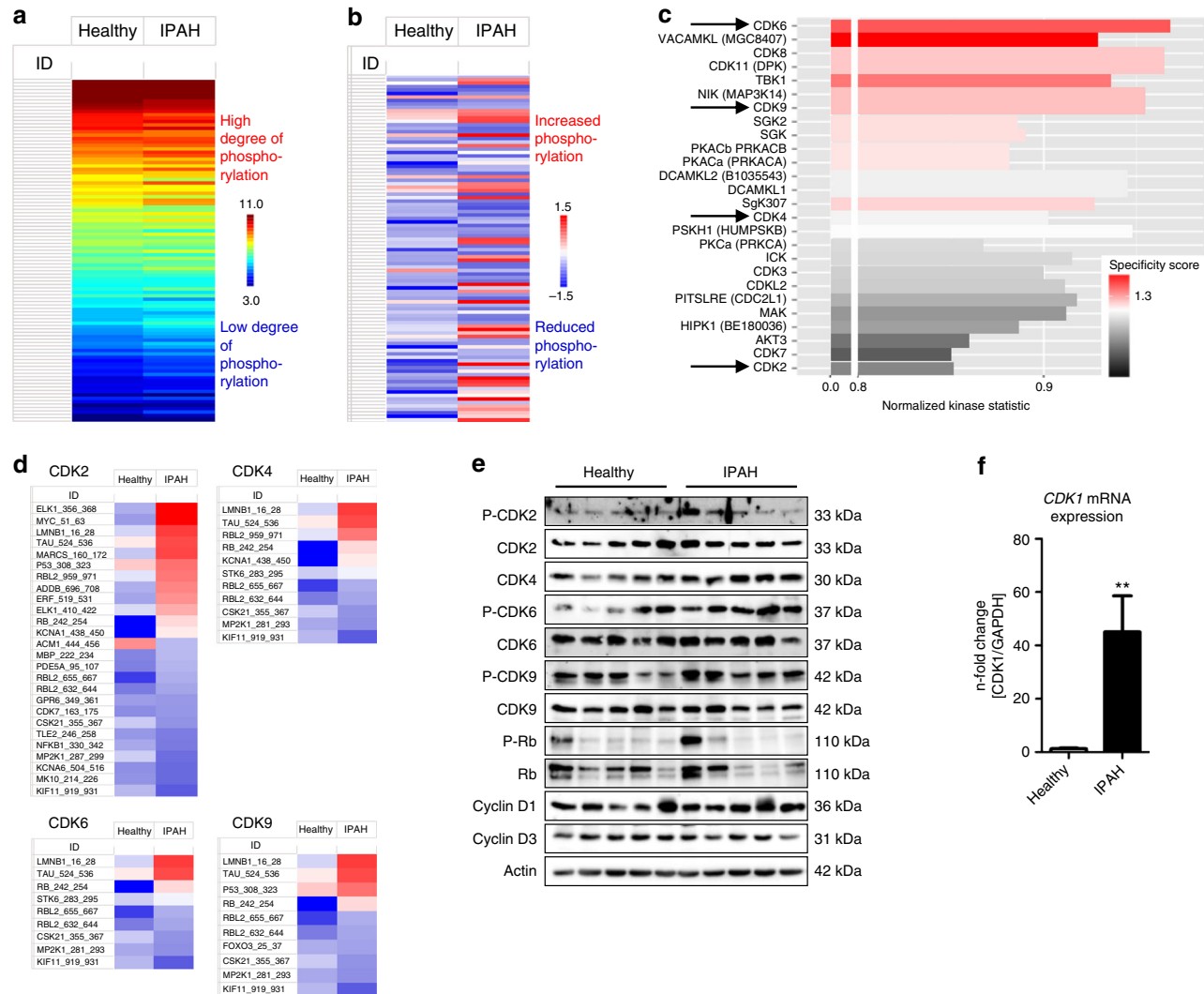

**Fig. 1** Kinome profiling reveals increased activity of the CDK-Rb-E2F signaling pathway in HPASMCs from IPAH patients. **a** Mean value of raw data for all individual samples, such as HPASMCs from healthy individuals ($n = 5$) and from IPAH patients ($n = 6$), presented as a heat map of log-transformed fluorescence signals upon substrate serine/threonine phosphorylation that are used to perform an upstream kinase analysis. **b** Mean value of a two-group comparison for all samples indicating the global changes in serine/threonine phosphorylation pattern between both HPASMCs entities under basal media conditions. **c** Bar chart displaying the overall prediction of kinases with higher activity in IPAH-patient-derived samples. Normalized kinase statistics is a mathematical-based algorithm indicating the estimated relative kinase activity while the specificity score reflects the reliability and accuracy of the prediction. **d** Two-group comparison for the four selected CDKs only including the phosphorylation pattern of their particular peptide substrate sets on the peptide array. **e** Western blot analysis for CDK2, CDK4, CDK6, and CDK9 activation and subsequent Rb-E2F downstream signaling in HPASMCs obtained from healthy donors and IPAH patients exposed to basal media. **f** CDK1 mRNA expression normalized to GAPDH as reference gene in HPASMCs of healthy individuals ($n = 3$) and IPAH patients ($n = 3$). After starvation, cells were cultured in basal media for 24 h. All data are presented as mean ± SEM of the $n$-fold change ($2^{-\Delta\Delta Ct}$) compared with a healthy control and analyzed statistically using a Mann–Whitney test; **$p < 0.01$. Source data are provided as a Source Data file

CDK6 (Fig. 2c), as well as for P-Rb and PCNA to demonstrate that the increased expression levels of those key proteins is associated with a phenotype of cell cycle dysregulation of diseased pulmonary vascular cells. These CDKs are expressed highly in explanted lungs from IPAH patients especially in pulmonary artery smooth muscle and endothelial cells, revealed by simultaneous staining with α-SMA and vWF. Adventitial fibroblasts and immune cells surrounding the obstructed vessels also stained positive for these proteins associated with an increased proliferation.

**Effects of CDK inhibitors on proliferation and CDK signaling.** Dinaciclib and palbociclib, two well-known CDK inhibitors, were administered at various concentrations in the presence of growth

medium (GM-2) to healthy HPASMCs and IPAH-HPASMCs to determine the $IC_{50}$ concentrations required to reduce cellular proliferation by half. Dinaciclib (Fig. 3a) exhibited a lower $IC_{50}$ value than did palbociclib (Fig. 3b) on healthy cells, which was further reduced in the case for IPAH-HPASMCs (Fig. 3c, d), indicating that disease-originating cells are more susceptible to targeted CDK inhibition. At the cellular level, increasing concentrations of both inhibitors (5 to 10 nM for dinaciclib; 0.1 to 1 μM for palbociclib) led to a dose-dependent decrease in P-Rb, Rb, and P-CDK2 expression levels (Fig. 3e, f). Clearly, CDK inhibition has a different mode of action compared to the tyrosine kinase inhibitor imatinib, the effect of which on IPAH smooth muscle cells is well-known: imatinib does not inhibit the Rb-E2F pathway (Fig. 3g). Both CDK inhibitors cause a subsequent

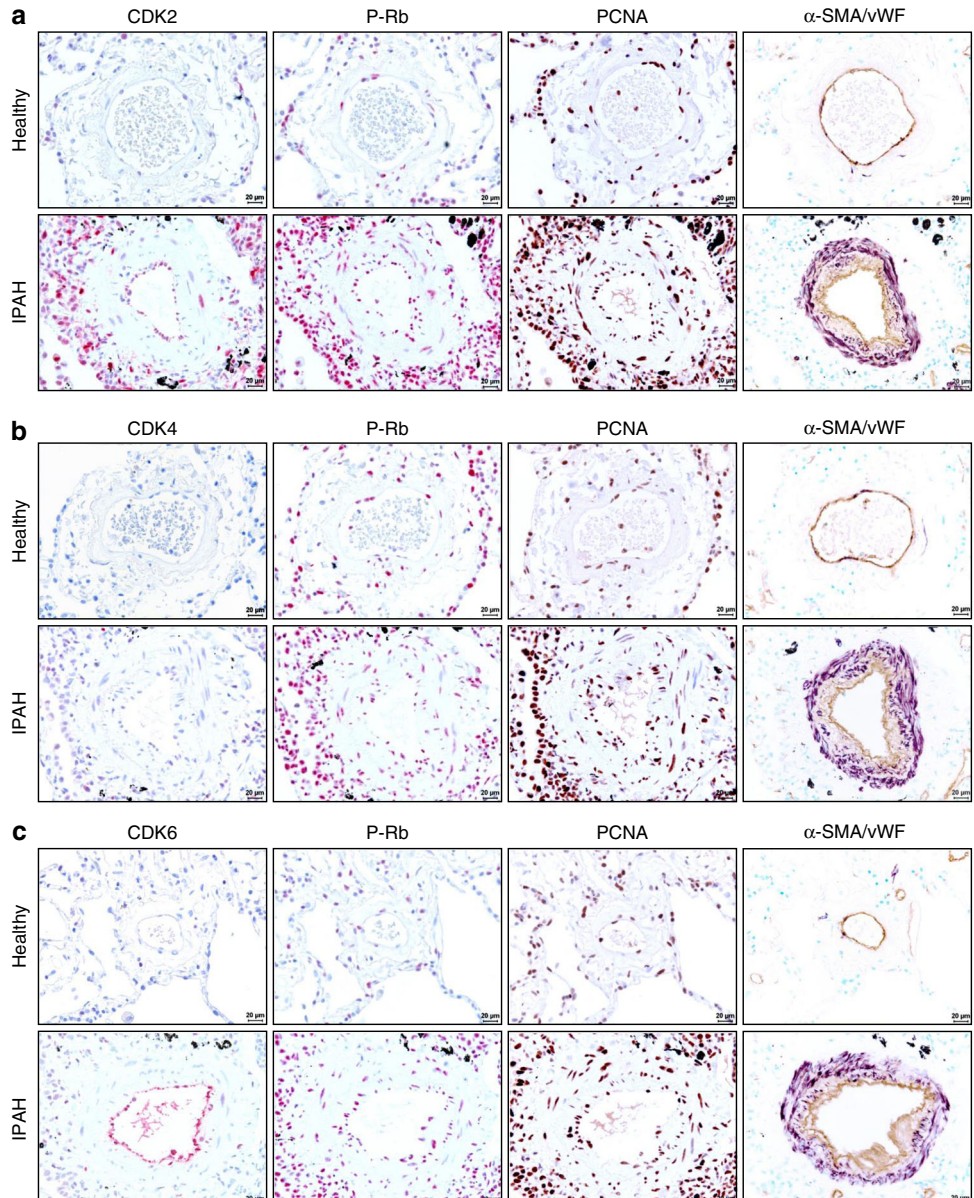

**Fig. 2** Immunohistological staining of distinct CDKs and P-Rb activation in human lung specimen. Representative images of serial sections from healthy ($n = 5$) and IPAH-patient derived lungs ($n = 5$) analyzed for CDK2 (**a**), CDK4 (**b**), and CDK6 (**c**), with corresponding IHC for P-Rb protein and its downstream target gene and common proliferation marker PCNA (proliferation cell nuclear antigen). Cellular identity was visualized by antibodies against α-SMA (alpha-smooth muscle actin) and vWF (von-Willebrand factor). Images were taken at 400-fold magnification with a scale bar of 20 μm

repression of E2F target gene expression (*CCNA2* and *CDK1*) (Fig. 3h, i). Interestingly, palbociclib (a CDK4/6 inhibitor) resulted in diminished (P-)CDK2 levels, which might be explained by the fact that a direct repression of CyclinA2 expression via Rb-E2F inactivation causes a decrease in the formation of active P-CDK2/CyclinA2 complexes[23,24]. Expression levels of other (P-)CDKs and cyclins remained unchanged. There was no sign of apoptosis detectable by an antibody specific for cleaved caspase-3.

**Analysis of the mode of action of dinaciclib and palbociclib.**
Healthy (Supplementary Fig. 4a) and IPAH- HPASMCs (Supplementary Fig. 4b) were treated with dinaciclib (CDK2 inhibitor) or palbociclib (CDK4/6 inhibitor) at different concentrations. An apparent reduction in proliferation was observed for dinaciclib at the low nanomolar range (5, 7.5, and 10 nM) after 24 h compared

with cells maintained in complete growth media with the vehicle DMSO (dimethyl sulfoxide). Dinaciclib reduced cell proliferation measured by BrdU incorporation in a concentration-dependent manner (Fig. 4a). Compared to cells treated with the positive controls staurosporine (0.2 μM SSP) and Triton X-100 (2%), no increase in lactate dehydrogenase (LDH) release was detected for any of the dinaciclib-treated cells (Fig. 4b); however, the overall metabolic activity and viability of the HPASMC was significantly suppressed after 48 h and 72 h (Fig. 4c), indicating low cytotoxicity. Similarly, palbociclib was tested at three different concentrations (0.1, 0.5, and 1 μM), which efficiently blocked DNA synthesis in a dose-dependent fashion (Fig. 4f) without increasing LDH release (Fig. 4g). In contrast to dinaciclib, palbociclib did not significantly reduce the metabolic activity and viability of HPASMCs after 72 h of treatment with the highest concentration (Fig. 4h). To provide evidence that the observed reduction in proliferation was directly linked to an arrest in cell cycle

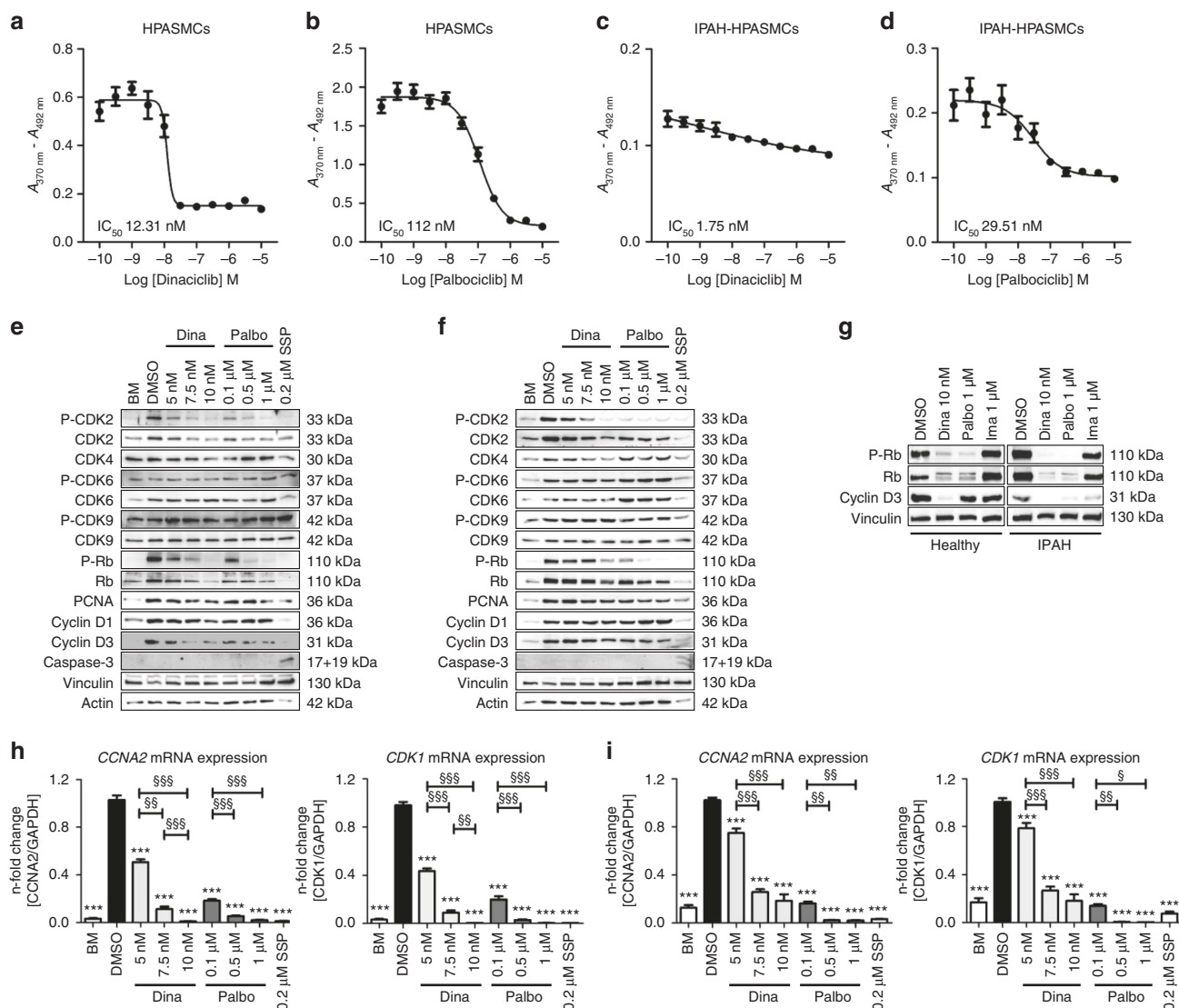

**Fig. 3** Evaluation of $IC_{50}$ concentrations of the CDK inhibitors dinaciclib and palbociclib on proliferation, and their effects on CDK-Rb-E2F signaling in human HPASMCs from healthy donors and IPAH patients. Cells were synchronized in basal media (BM) and then treated with the indicated concentrations of dinaciclib or palbociclib or DMSO as control in the presence of growth media (GM-2) for 24 h. $IC_{50}$ graphs determining the half maximal inhibitory concentration of dinaciclib (**a**, **c**) and palbociclib (**b**, **d**) to block GM-2 induced proliferation of healthy HPASMCs (**a**, **b**) and of IPAH-HPASMCs (**c**, **d**) as measured by BrdU incorporation (relative absorbance [$A_{370nm} - A_{492nm}$]). $IC_{50}$ values were calculated from measurements of two individual primary cell isolates (run twice in quartets) by the non-linear regression (curve fit) module with a variable slope using four parameters as provided by GraphPad Prism software. Representative western blots for the detailed analyses of cellular signaling on protein level after 24 h of treatment with dinaciclib, palbociclib, or staurosporine (SSP) in healthy HPASMCs (**e**) and IPAH-HPASMCs (**f**) with regard to CDK activation and downstream Rb-E2F pathway induction. **g** Western blot analysis for (P-)Rb and Cyclin D3 of healthy and IPAH-PASMCs treated with dinaciclib, palbociclib, or imatinib. *CCNA2* (left) and *CDK1* (right) mRNA expression (normalized to *GAPDH* as a housekeeping gene) of healthy HPASMCs (**h**) and diseased IPAH-HPASMCs (**i**) upon 24 h of inhibitor exposure. All data from two individual primary cell isolates (run twice in triplicates) are presented as mean ± SEM of the *n*-fold change ($2^{-\Delta\Delta Ct}$) compared with DMSO-treated control samples (black bar). Statistical analysis was performed using one-way ANOVA with Newman–Keuls post-hoc test for multiple comparisons; ***$p < 0.001$. *p*-values for distinct conditions are only given for their comparison with DMSO-treated control cells (black bar), and the significance of the difference between the three conditions with various concentrations of each of the CDK inhibitors is represented as follows: §$p < 0.05$, §§$p < 0.01$, §§§$p < 0.001$. Source data are provided as a Source Data file

progression, the cell cycle distribution of synchronized (starved) HPASMCs 24 h was assessed after incubation with dinaciclib (Fig. 4d, left) and palbociclib (Fig. 4i, left); as indicated in the two representative histograms. A concentration-dependent increase in the proportion of cells in the G1 phase after treatment with dinaciclib (Fig. 4d, right) or palbociclib (Fig. 4i, right) versus treatment with DMSO was noted. Simultaneously, the fraction of cells in the S and G2/M phase was decreased for both compounds, indicating a cell cycle arrest due to a disturbed progression from G1 to S phase. Furthermore, there were no signs of cell death induction, since the percentage of cells in the subG1 phase, which contains fragmented DNA from apoptotic cells, did not increase. A classical apoptosis detection assay was performed using annexin V (AV) and PI to rule out any direct cell death induction aside from the intended blockage of proliferation due to cell cycle arrest by the CDK inhibitors. Here, it became clear that dinaciclib actually triggers apoptosis-mediated cell death at the highest concentration of 10 nM. Fractions of $AV^+ PI^-$ (early apoptosis) as

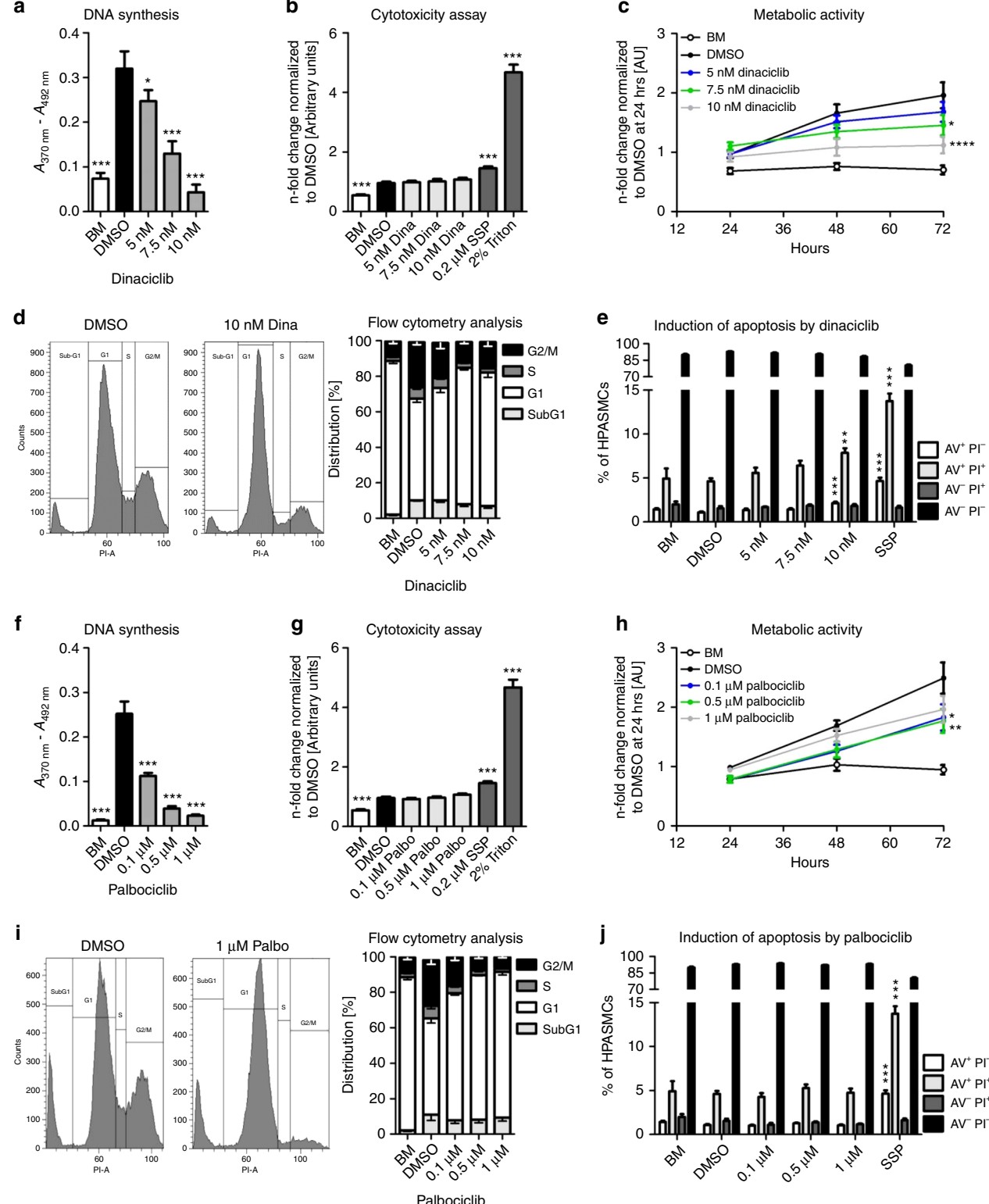

well as AV$^+$ PI$^+$ (late apoptosis) were clearly elevated almost to the same level as those measured in cells exposed to 0.2 μM staurosporine (SSP, positive control), when compared with healthy control cells (Fig. 4e). In contrast, palbociclib did not exhibit any pro-apoptotic effects (Fig. 4j). Dinaciclib and palbociclib both similarly affected the proliferation and cellular CDK-Rb-E2F-signaling of primary human pulmonary arterial endothelial cells (HPAECs). More precisely, HPAECs were starved in

endothelial-specific basal media supplemented with 0.5% FCS for 24 h prior to exposure to both CDK inhibitors in the presence of complete designated growth media. DNA synthesis assays revealed lower IC$_{50}$ values for dinaciclib (Supplementary Fig. 5a) than for palbociclib (Supplementary Fig. 5b). The doses applied for dinaciclib and palbociclib for former experiments were also applied to HPAECs to facilitate the comparison of the drug effects (Supplementary Fig. 5c). In the presence of both inhibitors a

**Fig. 4** Effects of the CDK inhibitors dinaciclib and palbociclib on proliferation, cell cycle, and apoptosis. HPASMCs were synchronized in BM and treated with dinaciclib (**a**–**e**), palbociclib (**f**–**j**), or DMSO (vehicle) in the presence of GM-2 for 24 h. **a**, **f** DNA synthesis was determined by measuring BrdU incorporation [$A_{370nm} - A_{492nm}$]. Data from four independent experiments performed in triplicates are presented as mean ± SEM. **b**, **g** Cytotoxicity was determined by detecting LDH release into the cell culture media [$A_{492nm} - A_{620nm}$]. Data from three independent experiments performed in up to six replicates are presented as mean ± SEM of the $n$-fold change normalized to DMSO-treated cells (black). Statistical analysis was performed using one-way ANOVA with Newman–Keuls post-hoc test for multiple comparisons; *$p < 0.05$, ***$p < 0.001$. $p$-values for distinct conditions are only given for their comparison with DMSO-treated control cells (black). **c**, **h** Metabolic activity was quantified up to 72 h during the treatments with dinaciclib (**c**) or palbociclib (**h**). Data from three independent experiments performed in triplicates are presented as mean ± SEM of the relative absorbance [$A_{492nm} - A_{690nm}$]. Statistical analysis was performed using two-way ANOVA (with repeated measures) with Bonferroni post-test for multiple comparisons versus DMSO-treated control cells; *$p < 0.05$, **$p < 0.01$, ****$p < 0.0001$. $p$-values are given for the time point of 72 h. Representative flow cytometry histograms upon PI labeling of DMSO-treated HPASMCs versus 10 nM dinaciclib (**d**, left) or 1 µM palbociclib (**i**, left) are shown. Four independent experiments were used to calculate the mean value for the SubG1 (light-gray), G1 (white), S (dark-gray), and G2/M (black) cell cycle phases upon dinaciclib (**d**, right) or palbociclib (**i**, right) incubation. Combined AV/PI staining was performed to discriminate between early (AV$^+$ PI$^-$, white) and late apoptosis (AV$^+$ PI$^+$, light-gray), dead (AV$^-$ PI$^+$, dark-gray), and living cells (AV$^-$ PI$^-$, black) due to dinaciclib (**e**) and palbociclib (**j**). Data from three independent experiments performed in triplicates are presented as mean ± SEM depicting the percentage of the indicated populations. Statistical analysis was performed using one-way ANOVA with Dunnett´s post-hoc test to compare all bars versus DMSO control; **$p < 0.01$, ***$p < 0.001$. Source data are provided as a Source Data file

decrease in protein levels of P-CDK2 and P-Rb, as well as PCNA (Supplementary Fig. 5d) was noted. Similarly, *CyclinA2* and *CDK1* exhibited a dose-dependent reduction in mRNA expression (Supplementary Fig. 5e). To demonstrate pulmonary selectivity of the CDK inhibitors, human aortic smooth muscle cells (HAoSMCs) were subjected to the same protocol as that was employed for HPASMCs. HAoSMCs were starved for 24 h in basal media without any source of growth factors or cytokines. Subsequently, cells were exposed to various concentrations of both inhibitors in the presence of standard growth media for 24 h. As illustrated in representative images of HAoSMCs treated either with dinaciclib (Supplementary Fig. 6a) or palbociclib (Supplementary Fig. 6d), neither of the CDK inhibitors affected cell density or morphology. In assays for LDH release (Supplementary Fig. 6b, e) and flow cytometric analysis (Supplementary Fig. 6c, f) for apoptosis induction, no signs of cell death were detectable upon CDK inhibition with concentrations ranging to 10 nM of dinaciclib and 1 µM of palbociclib compared with proper controls. In summary, it was concluded that neither dinaciclib nor palbociclib have any negative effects on the survival and viability of HAoSMCs from the systemic vasculature.

**Palbociclib improves cardio-pulmonary functions in MCT-induced PAH.** The MCT rat model of PAH was used further to elucidate whether CDK inhibition modulated pulmonary vascular remodeling and right ventricular (RV) function. In preliminary experiments, dinaciclib was administered at a daily dose of 7.5 mg/kg or 22.5 mg/kg body weight (BW) intraperitoneally from day 21 after MCT treatment, for a time period of 14 days. Owing to weight loss and severe side effects after 3 days, however, the study was discontinued as animals had to be relieved of their suffering. MCT-injected rats were also treated either with placebo or palbociclib at a dose 75 mg/kg BW from post-MCT day 21 for a time period of 14 days (Fig. 5a). The utility of palbociclib to improve the symptoms of P(A)H was demonstrated by improvements in right ventricular (RV) function. RV systolic pressure (RVSP) (Fig. 5b) and RV hypertrophy (Fig. 5f) were significantly altered in MCT rats treated with palbociclib compared with placebo. Moreover, CDK inhibition caused a significant decrease in RV dilatation (Fig. 5g), an increase in cardiac index (CI; Fig. 5c), and an increase in tricuspid annular plane systolic excursion (TAPSE; Fig. 5h), indicating an improvement in RV function. The hemodynamic benefits in response to palbociclib administration were accompanied by a decrease in the total pulmonary vascular resistance index (TPVRi; Fig. 5e). A comparison of selected factors pre- and post-treatment, including

stroke volume index (SVI), CI, RVID, TAPSE, is depicted together for the second animal model (Table 1). An improvement in pulmonary vascular remodeling (Supplementary Fig. 7a–c) was clearly demonstrated by a reduction in the percentage of fully muscularized small pulmonary arteries (Fig. 6a) and by a diminished medial wall thickness (Fig. 6b). These effects are a direct consequence of a targeted cell cycle arrest and subsequent inhibition of pulmonary vascular cell proliferation, as evident by a reduction in the proliferation index, the number of PCNA-positive cells (Fig. 6c). Western blot analyses clearly demonstrated reduced levels of (P-)CDK2, CDK6, Rb, PCNA, Cyclin D1, and Cyclin D3 (Fig. 6d). CDK4, P-CDK6, and P-Rb could not be reliably detected due to missing or unspecific binding of the antibodies tested. In line with this finding, at the molecular level, palbociclib almost completely reversed the MCT-associated upregulation of the two known CDK-Rb-E2F downstream targets, CCNA2 (Fig. 6e, left) and CDK1 (Fig. 6e, right), in lung homogenates.

**Palbociclib improves cardio-pulmonary functions in Su/Hox-induced PAH.** Similar to MCT-induced PAH, Su5416-injection and 3 weeks of hypoxia exposure followed by a time period of 2 weeks under normoxic conditions led to the onset of PAH. This could be antagonized with palbociclib resulting in a remarkable reduction in RVSP (Fig. 7b) and RV hypertrophy (Fig. 7f). RV function was almost normalized as demonstrated by improvements in RVID (Fig. 7g) and TAPSE (Fig. 7h). Pre- and post-treatment hemodynamic parameters are presented in Table 1, highlighting the in vivo efficacy of palbociclib. The impact of palbociclib results from the anti-remodeling effects of palbociclib on the pulmonary vasculature. Here, the process of pathological muscularization was significantly blocked, since the percentage of fully muscularized small arteries was appreciably decreased compared with non-treated Su/Hox animals. This decreased vessel muscularization was accompanied by an increase in the number of partially muscularized vessels (Fig. 8a). Both cell layers, the media and the neointima, were reduced in thickness upon palbociclib administration, indicating that the cell cycle of HPASMCs, as well as HPAECs (Supplementary Fig. 5) was blocked (Fig. 8b). This has a tremendous impact on the physical obstruction of vessels, where palbociclib led to an increase in the percentage of open vessels and a decrease in the number of closed arteries, as calculated by the occlusion score (Fig. 8c). Furthermore, the proliferation index of small pulmonary arteries was significantly lower in palbociclib treated Su/Hox rats than in the diseased animals (Fig. 8d). At the microscopic level, the in vivo

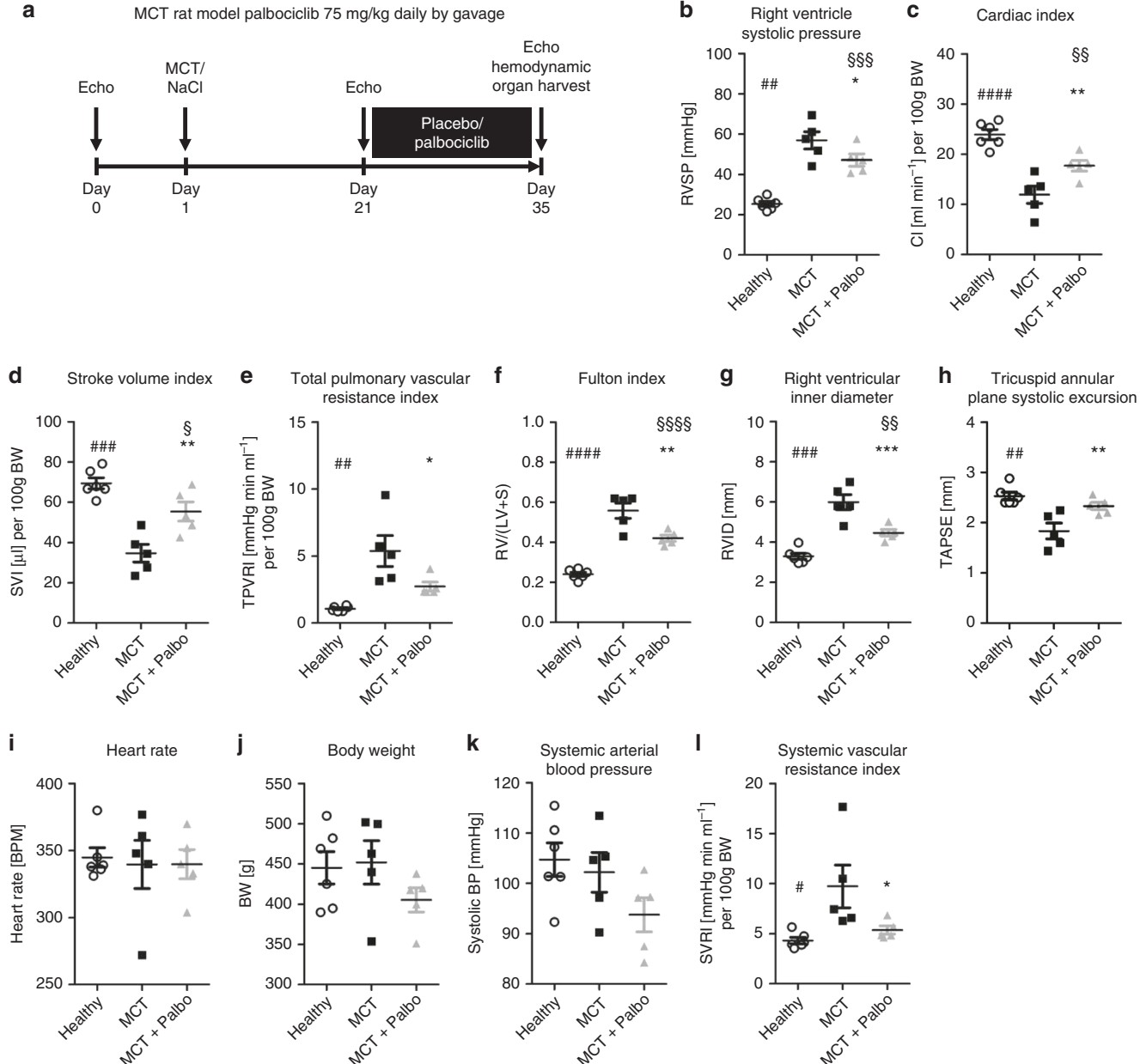

**Fig. 5** Effects of palbociclib on disease progression in the MCT rat model of pulmonary arterial hypertension. **a** Scheme of animal treatments. Echocardiograms were recorded at baseline, 21 days after MCT administration and again on day 35 after a 14-day period of treatment with 75 mg/kg body weight palbociclib (or placebo) daily by gavage. **b–l** On day 35, hemodynamics and cardiac function were assessed in vivo 14 days after treatment with palbociclib (MCT + Palbo, $n = 5$). Similarly, healthy rats (Healthy, $n = 6$) and animals injected with MCT and placebo (MCT, $n = 5$) were used as control groups. Data are presented as mean ± SEM and statistical analysis was performed using one-way ANOVA with Newman–Keuls post-hoc test for multiple comparisons; *$p < 0.05$, **$p < 0.01$, ***$p < 0.001$ for MCT + Palbo versus MCT; §$p < 0.05$, §§$p < 0.01$, §§§$p < 0.001$, §§§§$p < 0.0001$ for MCT + Palbo versus healthy; #$p < 0.05$, ##$p < 0.01$, ###$p < 0.001$, ####$p < 0.0001$ for healthy versus MCT. Source data are provided as a Source Data file

efficacy of palbociclib was noted from the reduction in the amount and severity of structurally remodeled vessels (Supplementary Fig. 8a–c). This phenomenon is a direct consequence of targeted CDK4 and CDK6 inhibition, revealing blockade of the downstream signaling via Rb-E2F at the protein level ((P-)CDK6, Rb, PCNA, CyclinD) (Fig. 8e), and at the level of *CyclinA2* and *CDK1* mRNA expression (Fig. 8f). Thus, the selective suppression of cell cycle progression achieved by inhibiting CDK with palbociclib appears to leads to an anti-remodeling process in small pulmonary arteries in both experimental models of PAH.

Toxicity and potential off-target effects are commonly encountered issues in the clinical application of kinase inhibitors not only

in the field of PAH[5]. In the case of dinaciclib, high mortality was noted in the MCT model, which led to a discontinuation of this animal study. Palbociclib is an approved anti-proliferative compound with a known safety profile, and no severe adverse events in terms of blood cell counts and organ damage were observed in the present study (Supplementary Figs. 9 and 10). Taken together, this demonstrates that palbociclib acts specifically on the hyperproliferative phenotype in the progression of PAH in the MCT-induced and in the Su/Hox rat model in vivo. CDK over-activation under these disease conditions was blocked efficiently by palbociclib, leading to a cell cycle arrest that finally interferes with structural remodeling processes in the pulmonary vasculature (Fig. 9).

**Table 1 Further hemodynamic description of both experimental PAH models**

| | MCT (n = 5) | MCT + palbociclib (n = 5) | Su/Hox (n = 8) | Su/Hox + palbociclib (n = 7) |
|---|---|---|---|---|
| BW [g] | 452 ± 30 (pre) | 420 ± 37 (pre) | 365 ± 20 (pre) | 363 ± 16 (pre) |
| | 451 ± 37 (post) | 405 ± 33 (post) | 391 ± 19 (post) | 362 ± 28 (post) |
| HR [bpm] | 342 ± 34 (pre) | 342 ± 39 (pre) | 349 ± 28 (pre) | 368 ± 16 (pre) |
| | 339 ± 40 (post) | 339 ± 24 (post) | 337 ± 17 (post) | 317 ± 42** (post) |
| SVI [µl] per 100 g BW | 46.8 ± 4.5 (pre) | 49.7 ± 7.7 (pre) | 35.5 ± 4.6 (pre) | 35.3 ± 3.2 (pre) |
| | 34.7 ± 9.9§ (post) | 55.4 ± 10.7## (post) | 33.3 ± 4.9 (post) | 54.9 ± 11.9****,#### (post) |
| CI [ml/min] per 100 g BW | 15.9 ± 1.9 (pre) | 16.7 ± 1.5 (pre) | 12.4 ± 1.5 (pre) | 12.8 ± 1.1 (pre) |
| | 11.9 ± 3.9§ (post) | 17.7 ± 2.4# (post) | 11.2 ± 1.7 (post) | 16.6 ± 3.5 **,### (post) |
| RVID [mm] | 4.2 ± 0.2 (pre) | 4.1 ± 0.3 (pre) | 4.7 ± 0.3 (pre) | 4.7 ± 0.3 (pre) |
| | 6.0 ± 0.8§§§§ (post) | 4.5 ± 0.4### (post) | 5.3 ± 0.5§§ (post) | 4.3 ± 0.5### (post) |
| TAPSE [mm] | 2.2 ± 0.2 (pre) | 2.4 ± 0.1 (pre) | 1.49 ± 0.07 (pre) | 1.46 ± 0.08 (pre) |
| | 1.8 ± 0.3§ (post) | 2.3 ± 0.2## (post) | 1.3 ± 0.1§§ (post) | 2.1 ± 0.1 ****,#### (post) |

Data, i.e., before (pre) and at the end of the treatment (post), are presented as mean ± SD and statistical analysis was performed using one-way ANOVA with Newman–Keuls post-hoc test for multiple comparisons. MCT rat model: $^§p < 0.05$, $^{§§§§}p < 0.0001$ for post-versus pre-treatment of MCT; $^#p < 0.05$, $^{##}p < 0.01$, $^{###}p < 0.001$ for post-treatment MCT + Palbociclib versus MCT. Su/Hox model: $^{§§}p < 0.01$ for post-versus pre-treatment of Su/Hox; $^{**}p < 0.01$, $^{****}p < 0.0001$ for post-versus pre-treatment of Su/Hox + Palbociclib; $^{###}p < 0.001$, $^{####}p < 0.0001$ for post-treatment Su/Hox + Palbociclib versus Su/Hox.
*BW* body weight, *HR* heart rate, *SVI* stroke volume index, *CI* cardiac index, *RVID* right ventricular internal diameter, *TAPSE* tricuspid annular plane systolic excursion. Source data are provided as a Source

## Discussion

The purpose of this study was to identify drug targets for the treatment of PAH by conducting a broad profiling screen and using functional assays in two animal models. Among the family of serine/threonine kinases, CDKs were found to be overactivated, using a peptide-based kinase activity assay to screen diseased HPASMCs, the essential pulmonary vascular cell type responsible for vessel obstruction in PAH. CDKs play essential roles in physiology and disease, making CDKs attractive but challenging drug targets[25]. CDKs have been reported to exhibit increased activation and expression in several cancer entities[10]. Accordingly, deregulated levels of members of both families of endogenous CDK inhibitors, namely INK and Cip/Kip, are found in a large variety of tumor types[26–28]. Cip/Kip family members like CDKN1A (p21) and CDKN1B (p27) have also been described as regulators of vascular smooth muscle cell proliferation[7,29,30]. Members of our research group have revealed that p21 is a central mediator between altered signaling pathways and proliferation, whereas successful anti-remodeling interventions lead to an upregulation of p21 expression[31,32]. These observations provide evidence for the similarity of signaling pathways that lead to uncontrolled cell growth in both cancer cells and PASMCs. We were thus encouraged to investigate the potential ability of known synthetic CDK inhibitors to substitute for the loss of endogenous CDK inhibitors and reverse the proliferative phenotype of pulmonary vascular cells in PAH.

Dinaciclib is a second-generation CDK inhibitor, which targets CDK1/2/5/9 and was evaluated for treating breast cancer, but undesirable side effects in part jeopardized its utility[33], most likely due to poor specificity, inappropriate selectivity[34] and potential off-target effects. Therefore, a third-generation selective CDK4/6 inhibitor, palbociclib, was included in the experimental protocols, which had previously exhibited a better risk/benefit ratio in breast cancer trials, and which has already been approved by the FDA[35,36]. Breast cancer patients often receive CDK4/6 inhibitors in combination with estrogen antagonists[37–39]. The therapeutic potential of both of these CDK inhibitors, dinaciclib and palbociclib (further reviewed in[40]) was examined for the treatment of PAH.

The expression profile of CDK2, CDK4, CDK6, and CDK9 was first assessed in specimens from IPAH patients and in animal models of PAH, namely hypoxia-induced PH in mice and MCT- as well as Su/Hox-induced PAH in rats. Although we focused primarily on the differential CDK activity in HPASMCs due to our initial data obtained by peptide-based kinase activity profiling, we could detect expression of CDK2, CDK4 and CDK6 also in other pulmonary vascular cell types (i.e., endothelial and immune cells, adventitial fibroblasts) in the vasculature of both donor and IPAH patients. However, increased immunoreactivity of the CDK targets P-Rb and PCNA was observed mainly in IPAH remodeled vasculature, suggesting increased CDK activity in IPAH is a prominent phenomenon driven by elevated secretion of growth factors from PASMCs or other vascular cells. Increased CDK activation in diseased IPAH-HPASMCs appears to be mainly due to elevated mRNA levels and not only due to higher levels of their regulating cyclin (as *CyclinA2* in the case for *CDK2*). This demonstrated that the hyperproliferative pathophysiological phenotype of pulmonary vascular cells, either in the whole-lung homogenates or in the isolated primary IPAH-HPASMCs, was driven by a loss of proper cell cycle control. Overexpression of CDK2 was confirmed in all three animal models, but only in MCT-treated rats CDK4 and CDK6 levels were elevated in the same manner. In both rat models, CDK6 was significantly upregulated under disease conditions. *CDK9* mRNA expression was enhanced in those animals suffering from PAH symptoms in all experimental models indicating also a higher activity of the transcriptional machinery.

These results concerning CDK expression supported further testing the effect of CDK inhibition in HPASMCs and in the MCT and Su/Hox models of PAH in vivo. To explore a potential means to slow down excessive cellular growth, which is the main finding in structurally remodeled small pulmonary arteries, dinaciclib and palbociclib were employed. In this regard, $IC_{50}$ doses for dinaciclib and palbociclib were determined using healthy HPASMCs and IPAH-HPASMCs. Here, diseased cells were demonstrably more susceptible to cell cycle arrest, based on reduced $IC_{50}$ values when compared with healthy HPASMCs. Dinaciclib and palbociclib were found to significantly reduce cellular proliferation by interfering with cell cycle progression from G1 into S phase in a concentration-dependent manner in HPASMCs. At the maximum concentration, dinaciclib increased apoptosis; however, in none of the western blots cleavage of caspase-3 was observed. In contrast, palbociclib appeared to be non-toxic. Detailed analysis of the signaling events in HPASMCs and HPAEC clearly confirmed that the proposed mode of action was consistent with CDK inhibition, as P-Rb levels and *CCNA2* and *CDK1* mRNA (CDK-Rb-E2F downstream target genes) are decreased. In human aortic smooth muscle cells, neither dinaciclib nor palbociclib exhibited any signs of cytotoxicity or induction of apoptosis and did not affect cell density and morphology.

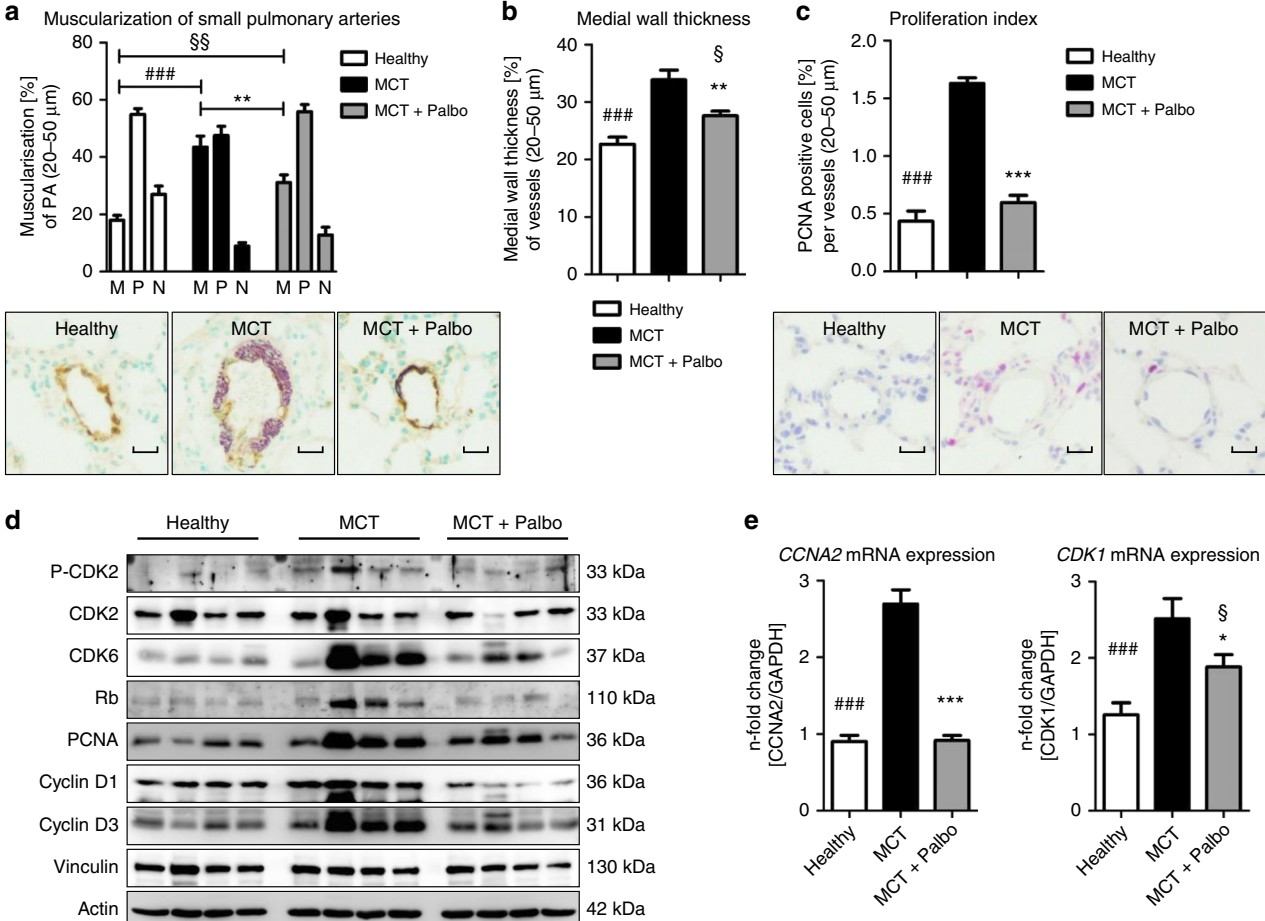

**Fig. 6** Ex vivo analyses of lung tissue for reversal of remodeling and in vivo drug efficacy in the MCT rat model. **a** The degree of muscularization of small pulmonary arteries (diameter 25–50 µm) was determined ex vivo via immunohistological staining of lung sections for vWF (brown) and α-SMA (violet) together with methylgreen for counterstaining. M: fully muscularized; P: partially muscularized; N: non-muscularized. Representative images for all three study groups are shown (healthy $n = 6$; MCT $n = 5$, MCT + Palbo $n = 5$). Images were taken at adequate magnification with a scale bar of 20 µm. **b** Medial wall thickness of vessels with a diameter of 20–50 µm was determined by Elastica-van-Gieson staining and is presented as percentage. **c** Proliferation index as a quantitative measure of PCNA-positive cells (purple) per vessel. Representative images (using hematoxylin/eosin as counterstain) for all three study groups are included. Images were taken at adequate magnification with a scale bar of 20 µm. Data from all individual animals (healthy $n = 6$; MCT $n = 5$, MCT + Palbo $n = 5$) are presented as mean ± SEM of 80–100 counted vessels and statistical analysis was performed using one-way ANOVA with Newman–Keuls post-hoc test for multiple comparisons; **$p < 0.01$, ***$p < 0.001$ for MCT + Palbo versus MCT; §$p < 0.05$, §§$p < 0.01$ for MCT + Palbo versus healthy; ###$p < 0.001$ for healthy versus MCT. **d** Western blot analysis for CDK2 and CDK6 activation and subsequent Rb-E2F downstream signaling in lung homogenates of representative samples from all three experimental groups (healthy, MCT, MCT + Palbo). **e** Analysis of *CCNA2* (left) and *CDK1* (right) mRNA expression normalized to *GAPDH* as reference gene in lung homogenates from all three experimental groups. Data from all individual animals are presented as mean ± SEM of the n-fold change ($2^{-\Delta\Delta Ct}$) compared with a healthy control rat (healthy $n = 6$; MCT $n = 5$, MCT + Palbo $n = 5$). Statistical analysis was performed using one-way ANOVA with Newman–Keuls post-hoc test for multiple comparisons; ***$p < 0.001$ for MCT + Palbo versus MCT; §$p < 0.05$ for MCT + Palbo versus healthy; ###$p < 0.001$ for healthy versus MCT. Source data are provided as a Source Data file

Both compounds are thus well-tolerated in this ex vivo setting and few negative side effects are to be expected for the systemic circulation.

Evaluation of the therapeutic potential in vivo was carried out in two experimental models for PAH (the MCT and the Su/Hox rat model). Both CDK inhibitors were tested in the MCT animal model, but the rats treated with dinaciclib exhibited severe side effects, which led to the discontinuation of its administration in vivo. It is possible that the animals were highly susceptible to dinaciclib due to the effects of the previously injected MCT[41–43]. In addition, it has been reported that inhibitors targeting CDK9 (of which dinaciclib is one example) typically block also other CDKs, thus promoting undesirable effects, such as dose-limiting toxicity in the bone marrow and the gastrointestinal tract in

clinical trials[44]. This may account, at least in part, for the adverse side effects noted for dinaciclib in the present study. Besides this, dinaciclib has a broader spectrum of activities as compared to palbociclib, which encompasses cell cycle inhibition, interference with protein synthesis (via CDK9[45–47]) and the regulation of the unfolded protein response (via CDK1 and CDK5[48]). This might explain the highly toxic profile of dinaciclib in vivo especially in combination with MCT. In contrast, palbociclib treatment in the MCT model, as well as in the Su/Hox model produced an impressive improvement in pulmonary and cardiac function as a consequence of directly interfering with pulmonary vascular remodeling and excessive proliferation. The increase in RV systolic pressure was reversed and the cardiac index was restored, which demonstrated a therapeutic effect of palbociclib in the

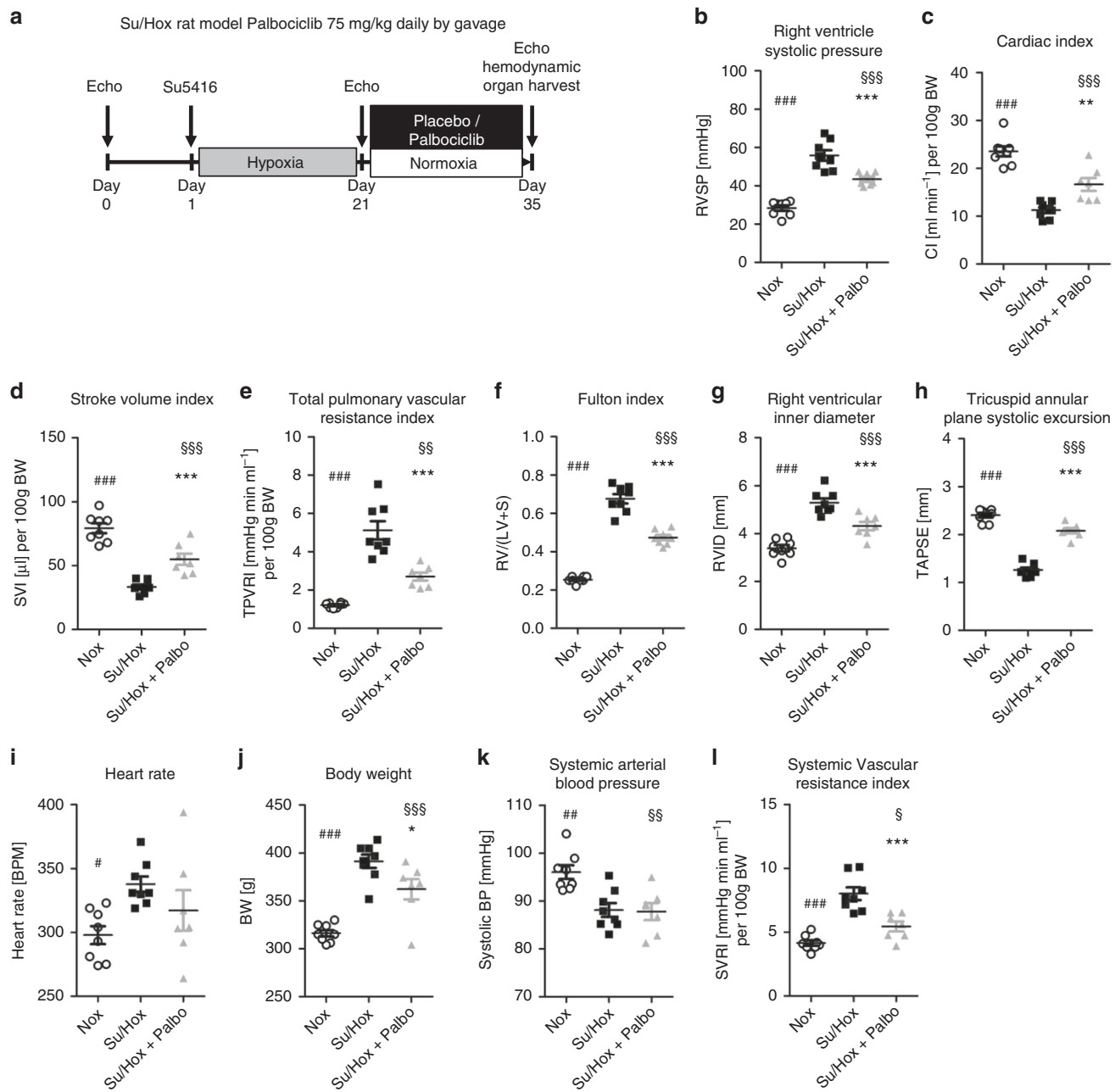

**Fig. 7** Effects of palbociclib on disease progression in the Su/Hox rat model of pulmonary arterial hypertension. **a** Scheme of animal treatments. Echocardiograms were recorded at baseline, 21 days after Su5416 administration at the beginning of hypoxia exposure and again on day 35 after a 14-day period of treatment with 75 mg/kg body weight palbociclib (or placebo) daily by gavage under re-exposure to normoxic conditions. **b–l** On day 35, hemodynamics and cardiac function were assessed in vivo 14 days after treatment with palbociclib (Su/Hox + Palbo, $n = 7$) (under initial hypoxia exposure). Similarly, normoxic rats ($n = 8$) and animals injected with Su5416 and placebo (under initial hypoxia exposure) (Su/Hox, $n = 8$) were used as control groups. Data are presented as mean ± SEM and statistical analysis was performed using one-way ANOVA with Newman–Keuls post-hoc test for multiple comparisons; *$p < 0.05$, **$p < 0.01$, ***$p < 0.001$ for Su/Hox + Palbo versus Su/Hox; §$< 0.05$, §§$p < 0.01$, §§§$p < 0.001$ for Su/Hox + Palbo versus Nox; #$< 0.05$, ##$p < 0.01$, ###$p < 0.001$ for Nox versus Su/Hox. Source data are provided as a Source Data file

absence of any adverse events. Palbociclib was tested in clinical trials in combination with the third-generation aromatase inhibitor letrozole[49] or with the selective estrogen receptor degrader fulvestrant[50] for the treatment of hormone receptor-positive/HER2-negative advanced metastatic breast cancer. The compounds were well-tolerated and the primary toxicity of asymptomatic neutropenia was effectively managed by dose modification, for example, by reduction, interruption, or treatment cycle delay without significant loss of efficacy. It has been shown that palbociclib only causes a reversible form of bone

marrow suppression, clearly differentiating it from apoptotic cell death caused by cytotoxic chemotherapeutic agents[51]. So far, palbociclib appears to display a comparably favorable safety profile. No signs of pulmonary vascular or cardiac toxicity or increased frequency of subdural hematoma were reported, which were partly limiting further investigation of kinase inhibitors such as dasatinib, sorafenib, sunitinib, and imatinib[5] in PAH.

Ex vivo analysis of the status of muscularization of small pulmonary arteries and of the molecular signaling pathway confirmed that palbociclib was acting according to the proposed mode of

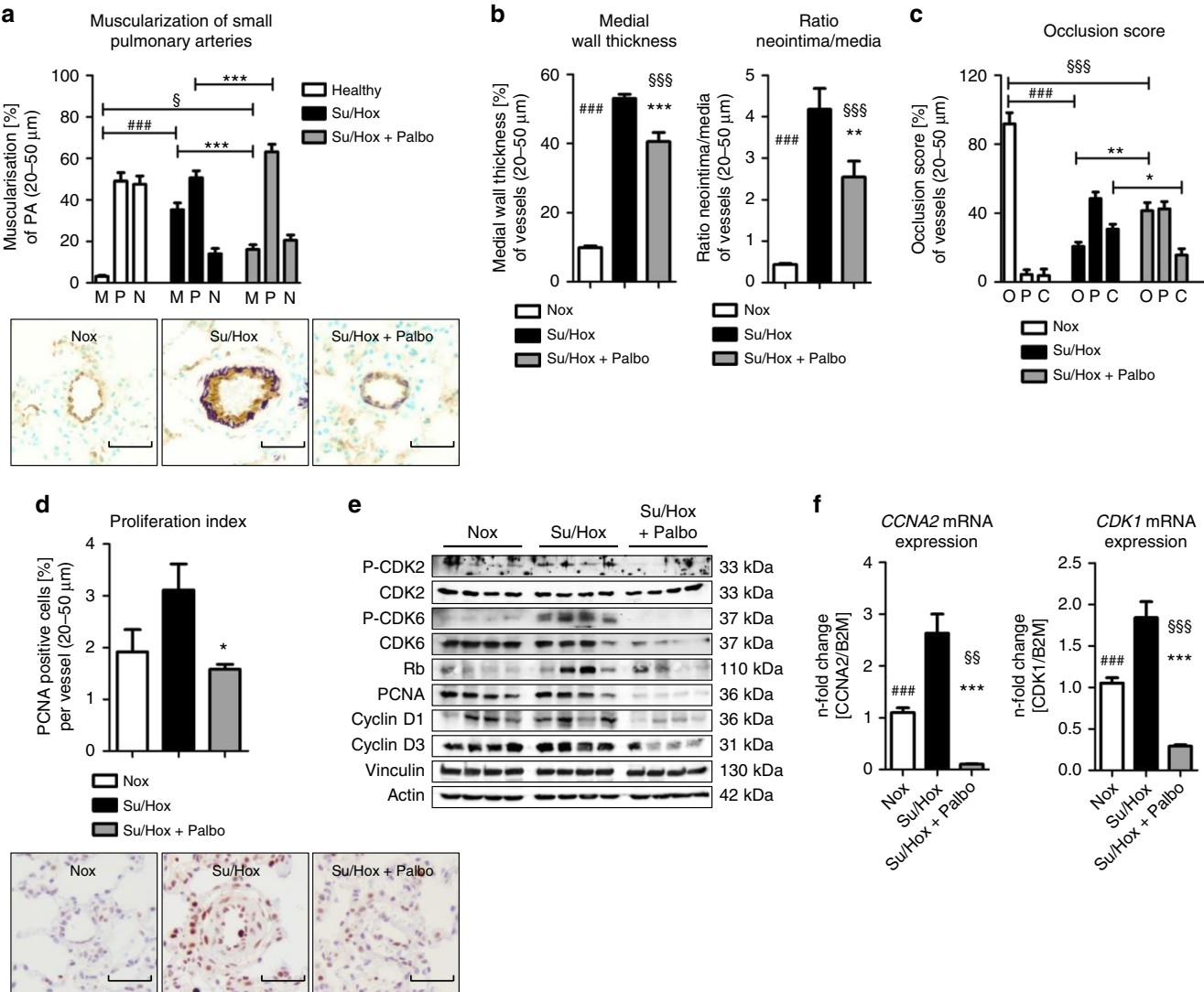

**Fig. 8** Ex vivo analyses of lung tissue for reversal of remodeling and in vivo drug efficacy in the Su/Hox model. **a** The degree of muscularization of small pulmonary arteries (diameter 25–50 μm) was determined ex vivo via immunhistological staining of lung sections for vWF (brown) and α-SMA (violet) together with methylgreen for counterstaining. M: fully muscularized; P: partially muscularized; N: non-muscularized. Representative images for all three study groups (Nox $n = 8$; Su/Hox $n = 8$; Su/Hox + Palbo $n = 7$) are shown. Images were taken at adequate magnification with a scale bar of 50 μm. **b** The medial wall thickness, as well as the corresponding ratio of neointima/media of the depicted study groups were determined by Elastica-van-Gieson staining allowing the calculation of the vessel occlusion score (**c**). O: opened; P: partial closed; C: closed. **d** Proliferation index as a quantitative parameter of PCNA-positive cells (purple) per vessel. Representative images (using hematoxylin/eosin as counterstain) for all three study groups are included. Images were taken at adequate magnification with a scale bar of 50 μm. Data from all individual animals (Nox $n = 8$; Su/Hox $n = 8$; Su/Hox + Palbo $n = 7$) are presented as mean ± SEM of 80–100 counted vessels and statistical analysis was performed using one-way ANOVA with Newman–Keuls post-hoc test for multiple comparisons; *$p < 0.05$; **$p < 0.01$, ***$p < 0.001$ for Su/Hox + Palbo versus Su/Hox; $^{\S}p < 0.05$; $^{\S\S\S}p < 0.001$ for Su/Hox + Palbo versus Nox; $^{\#\#\#}p < 0.001$ for Nox versus Su/Hox. **e** Western blot analysis for CDK2 and CDK6 activation and subsequent Rb-E2F downstream signaling in lung homogenates of representative samples from all three experimental groups (Nox, Su/Hox, Su/Hox + Palbo). **f** Analysis of *CCNA2* (left) and *CDK1* (right) mRNA expression normalized to *GAPDH* as reference gene in lung homogenates from all three experimental groups. Data from all individual animals are presented as mean ± SEM of the *n*-fold change ($2^{-\Delta\Delta Ct}$) compared with a healthy control rat (Nox $n = 8$; Su/Hox $n = 8$; Su/Hox + Palbo $n = 7$). Statistical analysis was performed using one-way ANOVA with Newman–Keuls post-hoc test for multiple comparisons; ***$p < 0.001$ for Su/Hox + Palbo versus Su/Hox; $^{\S\S}p < 0.01$, $^{\S\S\S}p < 0.001$ for Su/Hox + Palbo versus Nox; $^{\#\#\#}p < 0.001$ for Nox versus Su/Hox. Source data are provided as a Source Data file

action. Cellular proliferation was almost completely blocked in the animals treated with palbociclib, as evident by the reduced levels of CDK4, CDK6, Rb, Cyclin D1, and Cyclin D3 and of two important transcriptional CDK-Rb-E2F downstream target genes, *CCNA2* and *CDK1*. Although this analysis was performed in lung homogenates and not at the single-cell level, the influence on muscularization, namely, a reversal of structural remodeling, was evident as a decrease in the percentage of fully muscularized vessels, as well as a reduction in medial wall thickness, in the ratio

of neointima/media and in the occlusion score. Therapeutic approaches to treat PAH address pulmonary vasodilation as the main therapeutic target, rather than reversing the proliferative changes in the pulmonary circulation. However, anti-proliferative drugs have been examined for their utility in PAH, the most prominent example being imatinib. Imatinib, a PDGF receptor antagonist, is approved for the treatment of chronic myeloid leukemia and gastrointestinal stromal tumors, and leads to a regression of pulmonary vascular remodeling by PDGF inhibition

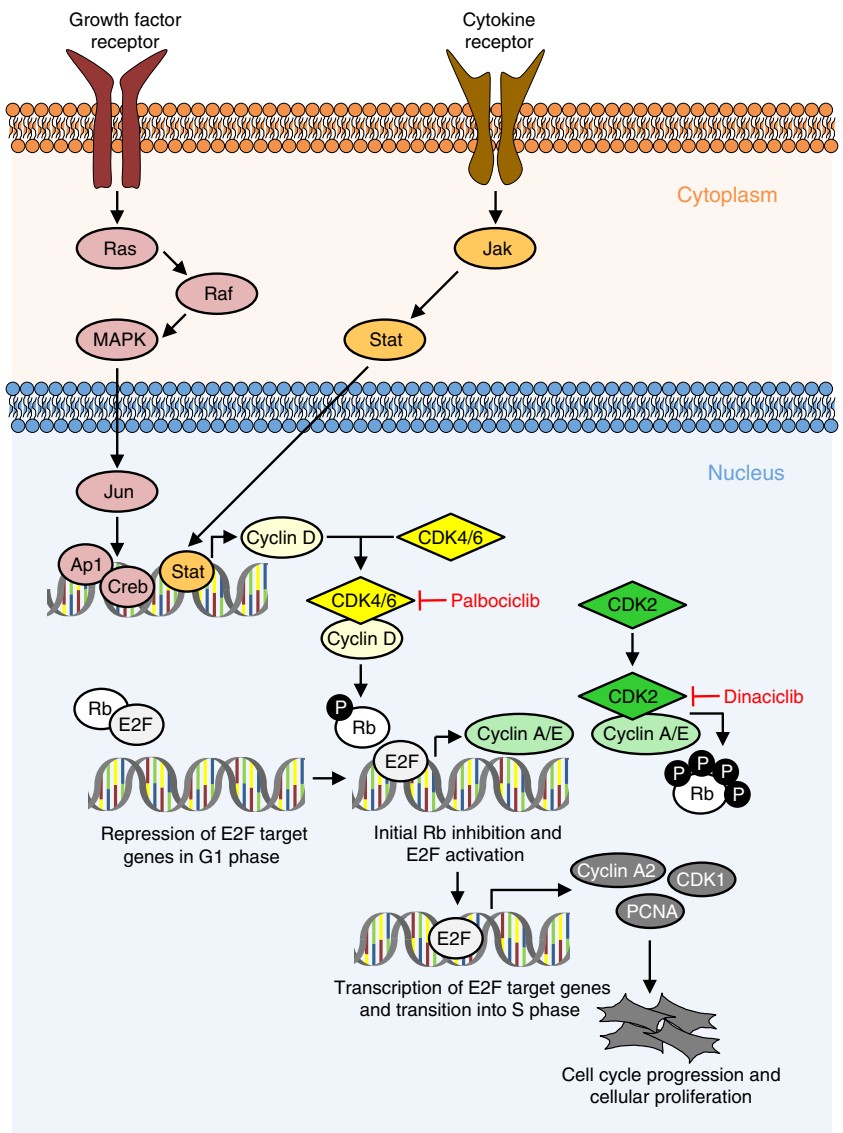

**Fig. 9** Proposed mechanism of action of palbociclib and dinaciclib in PAH. Multiple growth factors, cytokines, and mitogens induce the activation of cyclin-dependent kinases (CDKs), e.g., by increasing the expression of cyclin D1. Palbociclib specifically targets active complexes consisting of cyclin D1 and CDK4 or CDK6, whereas dinaciclib blocks the activity of the cyclin E-CDK2 complex. This leads to reduced levels of phosphorylated Rb protein and, as a consequence, transcriptional repression of E2F downstream target genes, e.g., *Cyclin A2*, *CDK1*, and *PCNA*. Thus, palbociclib (and dinaciclib) interfere with cell cycle progression from G1 into S phase, causing a reduction of proliferation of pulmonary vascular cells

in different animal models[52]. Clinical trials demonstrated that imatinib significantly improved exercise capacity and hemodynamics in patients with advanced PAH[53,54]. Several patients treated with imatinib, however, experienced subdural hematoma, which is a known, though rare, complication of this chemotherapeutic agent[55]. Subsequently, the benefit/risk ratio of imatinib was considered to be unfavorable and the imatinib was ultimately not approved for treating PAH patients.

Although monitoring of adverse effects will play an important role for future evaluation of palbociclib as a possible treatment option in PAH, experiences from the field of oncology are so far encouraging. Neutropenia, known to be an undesirable side effect of palbociclib and potentially limiting a long-term clinical use, could be circumvented by dose adaptation (as previously mentioned), combined therapy with other approved PAH drugs and/or by local administration (i.e., inhalation). As reported for cancer cells and demonstrated by our data, this small-molecule inhibitor exhibits strong selectivity for CDK4 and CDK6 with potent anti-

proliferative effects. The strong anti-proliferative profile of this approved CDK inhibitor suggests that this molecule has the necessary properties to allow further investigation of its therapeutic benefit in PAH patients.

## Methods

**Cell culture**. Human pulmonary arterial smooth muscle cells (HPASMCs) from healthy individuals were obtained from Lonza Ltd. (Basel, Switzerland), as well as from the UGMLC Giessen Biobank. HPASMCs from IPAH patients were obtained exclusively from the UGMLC Giessen Biobank. The Ethics Committee of the Justus Liebig University has approved the biomaterial collection of the UGMLC/DZL biobank under the ethics vote number 58/15. The patients have been informed and given their written consent for the use of biomaterials for research purposes. All studies and procedures to obtain human specimen were conducted according to the Declaration of Helsinki. Human aortic smooth muscle cells (HAoSMCs) were also purchased from Lonza. HPASMCs and HAoSMCs were maintained in culture with smooth muscle growth medium-2 (GM-2; Lonza) in a humidified incubator at 37 °C aerated with 5% $CO_2$ in air. The GM-2 contained FBS, hFGF-B, hEGF, insulin, and gentamicin/amphotericin-B. All experiments were performed with cells between passages 4 and 6. Cells were seeded on different types of dishes

according to the different assays. Six hours after seeding, cells were starved for 18 h by using smooth muscle basal medium (BM; Lonza) containing no additional growth factors, cytokines, or supplements before subsequent treatments. Dinaciclib (SCH727965), palbociclib-HCl (PD-0332991), and imatinib (STI571) were purchased from Selleckchem by the Absource Diagnostics GmbH (Munich, Germany). Staurosporine was obtained from Merck Chemicals GmbH (Darmstadt, Germany). Cells were exposed to inhibitors at concentrations stated in the figures for 24–72 h in the presence of GM-2. Visual analysis was carried out by bright-field microscopy (Eclipse TS100; Nikon GmbH, Duesseldorf, Germany) at magnifications stated in the figure legends.

**Protein lysate for the peptide-based kinase activity assay**. For protein isolation, cell culture dishes were placed on ice and the medium was aspirated. The cells were washed twice with 5 ml ice-cold PBS. Cells were scraped from the dishes in 200 µl of M-PER lysis buffer (Thermo Fisher Scientific, Waltham, MA, USA) containing protease and phosphatase inhibitor cocktails (Pierce, Rockford, IL, USA). The lysate was transferred to a pre-cooled tube and incubated for 1 h at 4 °C with constant agitation; this was followed by centrifugation at $16,000 \times g$ for 15 min at 4 °C. The supernatant was divided into 15-µl aliquots that were immediately flash-frozen in liquid nitrogen and stored at −80 °C. Protein concentration was determined using a bicinchoninic acid (BCA) protein assay kit (Thermo Fisher Scientific) according to manufacturer's instructions. Aliquots were used for primary testing by western blot analyses and subsequently applied in the peptide-based kinase activity assay.

**Peptide-based kinase activity assay**. Assessment of the peptide-based kinase activity was conducted using the PamStation®12 platform (PamGene International, s-Hertogenbosch, Netherlands) with Evolve 12 software. A total of 1 µg of protein lysate was dispensed onto an array of the PamChip STK (serine/threonine kinase) dissolved in protein kinase buffer (1x PK), including 1% BSA, 0.46 µl STK antibody mix and 400 µM ATP (provided by PamGene). Detection of substrate phosphorylation was enabled by a FITC-conjugated secondary phospho-S/T-specific antibody, which was applied in a subsequent step with corresponding antibody buffer. Measurement as well as image acquisition and data analysis were performed according to the manufacturer's instructions. The list of significantly deregulated peptides showing differential patterns of phosphorylation was generated by a comparison of lysates of HPASMCs obtained from healthy individuals with those from IPAH patients cultured under basal media conditions (BM). Data and upstream kinase analysis was conducted using the Bionavigator software v. 6.2 (PamGene) as further described in the supplementary material.

**Western blot analysis**. If not mentioned otherwise, as in the case for the M-PER lysates, protein samples of HPASMCs and lung tissue were extracted after lysing cells in RIPA lysis and extraction buffer containing protease and phosphatase inhibitor cocktails (Thermo Fisher Scientific). Western blot analysis was performed using standard techniques for all lysates, regardless of the previously mentioned isolation techniques. Equal amounts of total protein (30 µg or up to 80 µg in the case of lung homogenates from rats) were separated by SDS polyacrylamide gel electrophoresis and transferred to nitrocellulose membranes for 60 min at 100 V. Membranes were blocked for 1 h at room temperature in 5% non-fat dry milk dissolved in 1% Tween-20/TBS (T-TBS). After blocking, membranes were incubated with the primary antibody at 4 °C overnight in either T-TBS with 5% BSA or in T-TBS with 5% non-fat dry milk, according to the manufacturer's instructions for the respective antibody. Primary antibodies against cleaved Caspase-3 (#9665, 1:500 dilution), (P-)CDK2 (#2561, #2546, 1:500–1:1000 dilution), CDK4 (#12790, 1:500–1:1000 dilution), CDK6 (#3136, 1:1000–1:2000 dilution), (P-)CDK9 (#2549 S, #2316 S, 1:1000 dilution), cyclin D1 (#2978, 1:500–1:1000 dilution), cyclin D3 (#2936, 1:1000 dilution), GAPDH (#2118, 1:2000 dilution), P-ERK (#4370, 1:1000 dilution), (P-)Rb (#8516 S, #9309 S, 1:1000–1:2000 dilution) were obtained from Cell Signaling Technology (Danvers, MA, USA), for P-CDK6 (#ab131439, 1:500 dilution), beta-actin (#ab8226, #ab8227, 1:2000 dilution) and vinculin (ab18058, 1:2000 dilution) from Abcam (Cambridge, UK), and for ERK (#sc-93, 1:2000 dilution), and PCNA (#sc-56, 1:500–1:1000) from Santa Cruz Biotechnology (Heidelberg, Germany). After washing with T-TBS, the HRP-labeled secondary antibody (#7074S, #7076S, Cell Signaling Technology) was diluted 1:2000 in T-TBS with 5% non-fat dry milk and incubated with the membrane for 90 min at room temperature. After washing with T-TBS, protein bands were visualized using Super Signal West Femto Maximum Sensitivity Substrate (Thermo Fisher Scientific) or Amersham ECL Prime Western Blot Detection Resolution (GE Healthcare Europe GmbH, Freiburg, Germany) and images were captured by a ChemoCam Imager device from Intas Science Imaging Instruments GmbH (Göttingen, Germany) using the ChemoStar Imager software. All uncropped scans are supplied in the corresponding source data file.

**RNA isolation, cDNA synthesis, and real-time PCR**. Total RNA from cultured cells was isolated 24 h after indicated treatments and purified with on-column DNase digestion using miRNeasy Mini Kit (Qiagen GmbH, Hilden, Germany) according to the manufacturer's protocol. Human lungs from which total RNA was isolated were obtained from Hannover Medical school. The study was designed and

performed following the requirements of the local ethics committee at Hannover Medical school (ethics vote no. 2702–2015); the patients included consented in writing to the use of their biomaterials for research purposes. RNA integrity was verified for representative samples from different species on agarose gel electrophoresis (Supplementary Fig. 11) and by a Bioanalyzer instrument (Agilent Technologies, Santa Clara, CA, USA). Reverse transcription was performed using iScript cDNA Synthesis Kit (Bio-Rad Laboratories GmbH, Munich, Germany) and by incubating the complete reaction mix, including 1 µg RNA in a thermocycler using the following conditions: 5 min at 25 °C, 20 min at 46 °C, 1 min at 95 °C, and holding at 4 °C. The resulting cDNA was diluted twofold with RNA/DNAse-free water. Real-time PCR reactions were set-up in a 96-well format containing iTaq Universal SYBR Green Supermix (Bio-Rad Laboratories GmbH, Munich, Germany), 500 nM of each of forward and reverse primer (Metabion International AG, Planegg/Steinkirchen, Germany), and 1–6 µl of diluted cDNA sample and $H_2O$ in a 20-µl reaction volume. Real-time fluorescence of PCR products was detected using a Mx3000P qPCR System (Agilent Technologies, Santa Clara, CA, USA) using the following thermocycling conditions: 1 cycle of 95 °C for 30 s (first segment); 40 cycles of 95 °C for 10 s, and 58 °C for 45 s (second segment); 1 cycle of 95 °C for 1 min and 58 °C for 30 s followed by 0.5 °C increments at 2 s/step back to 95 °C for 30 s (third segment). Each gene was normalized to a housekeeping control as indicated. Specificity of the primer pairs was evaluated using agarose gel electrophoresis and Bio-Rad GelDoc XR + system (Bio-Rad Laboratories GmbH, Munich, Germany). Only primer pairs leading to the synthesis of a single fragment with the appropriate size were used in this study. Primer sets used for this study are listed in the supplementary information (Supplementary Table 1).

**Preparation and immunohistology of explanted human lungs**. Patients were diagnosed for PAH according to international guidelines. Peripheral lung tissue samples were obtained from the UGMLC Giessen Biobank. The Ethics Committee of the Justus Liebig University has approved the biomaterial collection of the UGMLC/DZL biobank under the ethics vote number 58/15. The patients have been informed and given their written consent for the use of biomaterials for research purposes. All studies and procedures to obtain human specimen were conducted according to the Declaration of Helsinki. Samples were placed in 4% paraformaldehyde (PFA) immediately after explantation and further embedded in paraffin. Serial sections (2 µm) were incubated with antibodies specific for CDK2, CDK4, CDK6, P-Rb, PCNA, α-SMA, and vWF to identify cells positive for the given protein of interest. All experimental details can be found in the section of the supplementary material.

**Cytotoxicity, DNA synthesis, and metabolic activity assays**. Kits for detection of cytotoxicity (LDH), cell proliferation via ELISA (BrdU, colorimetric), and cell proliferation kit II (XTT) were purchased from Roche Diagnostics GmbH (Mannheim, Germany) and performed according to the manufacturer's instructions. Briefly, cultured HPASMCs were starved for 18 h in the presence BM before treatment with dinaciclib, palbociclib, staurosporine (Merck Chemicals GmbH, Darmstadt, Germany), Triton, or DMSO at the given concentrations for 24 h. Cell culture supernatants were collected and centrifuged, and enzymatic activity of LDH was determined in a colorimetric reaction. Quantification was carried out by absorbance measurement at 492 nm using 620 nm as a reference wavelength. Final graphs represent results from three individual experiments performed in triplicate (with repeated determination), and data are given as an $n$-fold change normalized to the absorbance for DMSO-treated control cells. For investigations of DNA synthesis and metabolic activity to determine the proliferation rate and cellular viability, 4000 HPASMCs were seeded in 96-well microtiter plates in a volume of 100 µl per well. Six hours later cells were starved for 18 h by BM before subsequent treatment with dinaciclib, palbociclib, or DMSO at given concentrations for 24 h; metabolic activity was investigated also after 48 and 72 h. Quantification of DNA synthesis by a colorimetric BrdU ELISA was carried out by absorbance measurement at 370 nm using 492 nm as a reference wavelength and final graphs represent data from three individual experiments performed in triplicates. For the metabolic XTT assay, detection was carried out at 492 nm using 690 nm as a reference wavelength. Results are given at an $n$-fold change normalized to the absorbance for DMSO-treated control cells after 24 h. The Infinite M200 Pro instrument from Tecan Group (Männedorf, Switzerland) was used to determine the absorbance at the desired wavelengths.

**Cell cycle analysis**. As described above, cells were cultured in 10-cm dishes and starved for 18 h in BM. Treatment with CDK inhibitors was performed in the presence of GM-2 for 24 h. Analysis of the cell cycle was performed as described by others[56]. Briefly, cells and conditioned media were collected by trysinization followed by centrifugation at $600 \times g$ for 5 min at 4 °C with brakes disabled; pellets were washed twice in ice-cold PBS and fixed overnight in 70% ethanol at 4 °C. Thereafter, cells were washed in 38 mM sodium citrate buffer (pH 7.4) and incubated in 300 µl hypotonic DNA staining solution (0.05 mg/ml PI, 5 µg/ml RNase A and 38 nM sodium citrate) at room temperature for 30 min protected from light. Flow cytometric measurement was carried out with a BD FACSCANTO II flow cytometer. Analysis of the different cell cycle phases (Sub-G1, G1, S, and G2/M) was performed with BD FACSDiva software (Version 6.1.3), and mean values from

four individual sets of experiments were recorded. A figure exemplifying the gating strategy is provided in the supplementary information (Supplementary Fig. 12a–c).

**Apoptosis detection assay**. Detailed analysis of apoptosis-mediated cell death was performed by classical annexin V (AV) staining combined with propidium iodid (PI) using the FITC Annexin V Apoptosis Detection Kit I (BD Bioscience, Heidelberg, Germany) according to the manufacturer's guidelines. In summary, as in the conditions used for the cell cycle analyses, HPASMCs or HAoSMCs were seeded in triplicate, starved, and treated for 24 h with the different compounds. For sample preparation, cells including cell culture supernatant were collected by accutase digestion (PAN-Biotech GmbH, Aidenbach, Germany), centrifuged at $600 \times g$ for 5 min at 4 °C with brakes disabled, and carefully washed twice with cold PBS. Cell pellets were resuspended in 200 µl AV binding buffer to a concentration of $10^6$ cells per ml, from which 100 µl were stained with 5 µl of FITC-conjugated AV-FITC and 5 µl PI solution. Cells were incubated for 15 min at room temperature in the dark and further diluted with 100 µl AV binding buffer prior to being assessed within 1 h by flow cytometry with detection of PI in the PerCP-Cy5.5 channel. A figure exemplifying the gating strategy is provided in the supplementary information (Supplementary Fig. 12d–f). Data from three independent experiments performed in triplicate were analyzed by FlowJo software (Version 10) and are summarized in one graph.

**In vivo study design**. The experiments were performed in accordance with the US National Institutes of Health Guidelines on the Use of Laboratory Animals. Both the University Animal Care Committee and the federal authorities for animal research of the Regierungspräsidium Giessen and Darmstadt (Hesse, Germany) approved the study protocol (approval numbers V 54–19 c 20 15h 01 GI 20/10 Nr. G 50/2016 and V54–19c 20/15-B2/1195). Rats (Sprague-Dawley, male, 300–350 g body weight) were subcutaneously injected with MCT (60 mg MCT per kg body weight) and randomized on day 21 to receive a daily dose of palbociclib (75 mg per kg body weight) or vehicle by oral gavage during the subsequent 2 weeks. Rats injected with saline instead of MCT on day 1 were used as a healthy control group. For the SU5416-hypoxia rat model, Wistar-Kyoto rats (male, 300–350 g body weight) were injected with SU5416 (20 mg/kg body weight) and kept in hypoxic chambers for 3 weeks where oxygen levels were reduced to 10%. Animals were then re-exposed to normoxic conditions for 2 additional weeks and a daily dose of palbociclib (75 mg per kg body weight) or vehicle was administered by gavage.

**Echocardiography**. Transthoracic echocardiography was performed with a VEVO2100 (Visualsonics, Toronto, Canada) system equipped with a 13–24-MHz transducer (MS250, rat cardiovascular) to measure right ventricular internal diameter (RVID), tricuspid annular plane systolic excursion (TAPSE), and cardiac output (CO). Briefly, anesthesia was induced with isoflurane gas (3%) and maintained with 1.5% isoflurane in room air supplemented with 100% $O_2$. Rats were positioned a supine position on a heating platform with all legs taped to ECG electrodes for heart rate (HR) monitoring. Body temperature was monitored via a rectal thermometer (Indus Instruments, Houston, TX). The chest of the rats was shaved and treated with a chemical hair remover to reduce ultrasound attenuation. To provide a coupling medium for the transducer, a pre-warmed ultrasound gel was spread over the chest wall. RVID was measured as the maximal distance from the RV free wall to the septum using the apical four-chamber view. To determine TAPSE, the M-mode cursor was oriented to the junction of the tricuspid valve plane and the RV free wall using the apical four-chamber view. PA diameter was measured at the level of the pulmonary outflow tract during midsystole using the superior angulation of the parasternal short-axis view. Pulsed-wave Doppler was used to measure the PA flow velocity-time integral (VTI). By combining PA VTI, pulmonary artery area, and heart rate, a value for CO was derived echocardiographically. The total pulmonary vascular resistance index was calculated using the following formula: TPVRI = RVSP/CI, where RVSP is the right ventricular systolic pressure (mm Hg) and the cardiac index (CI in ml/min per 100 g body weight) is the cardiac output (CO in ml/min) normalized to 100 g body weight. SVI is defined by SAP/CI.

**Hemodynamic measurements**. Measurement of right ventricular systemic pressure (RVSP) and systemic arterial pressure (SAP) were performed under anesthesia. After tracheotomy the rat was placed supine on a homeothermic plate (AD Instruments, Spechbach, Germany) and connected to a small-animal ventilator MiniVent type 845 (Hugo Sachs Elektronik, March-Hugstetten, Germany). The body temperature was controlled by the rectal probe connected to the control unit (AD Instruments, Spechbach, Germany) and was maintained at 37 °C during the catheterization. The right external jugular vein was catheterized with a high-fidelity 1.4F micromanometer catheter (Millar Instruments, Houston, USA) and advanced into the right ventricle to assess RVSP. Data were collected and analyzed using the PowerLab data acquisition system (MPVS-Ultra Single Segment Foundation System, AD Instruments, Spechbach, Germany) and LabChart 7 for Windows software.

**Histomorphometry of explanted lungs**. After exsanguination, the left lung was fixed for histology in 3.5–3.7% PFA and the right lung was snap-frozen in liquid nitrogen. For right heart hypertrophy, the right ventricle (RV) was separated from the left ventricle plus septum (LV+S), and the RV/(LV+S) ratio was determined from the tissue. For pulmonary vascular morphometry, lungs were flushed with saline at a vascular pressure of 22 cm $H_2O$. The left lung was stored in 3.5–3.7% PFA for the next 24 h and then in PBS before being dehydrated. The lung was then embedded in paraffin and 3-µm thick samples were generated with a microtome. The degree of muscularization of small peripheral pulmonary arteries was assessed by double-staining the sections with an anti-α-smooth muscle cell actin antibody (dilution 1:700, clone 1A4, Sigma-Aldrich now Merck, Darmstadt, Germany) and an anti-human von-Willebrand factor antibody (dilution 1:2000, Dako, Hamburg, Germany). Stained thin sections were examined by light microscopy, and the color along the perimeter of the vessel was analyzed using a computerized morphometric system (Qwin, Leica, Wetzlar, Germany) that differentiates the purple staining of the smooth muscle and the brown staining of the endothelial layer. At 400-fold magnification, 80–100 intra-acinar vessels accompanying either alveolar ducts or alveoli of 25 to 50 µm were analyzed by an observer blinded to the treatment of the animals. Each vessel was categorized as either non-muscularized (5% SMC actin around the vessel), partially muscularized (5 to 75% SMC actin around the vessel) or fully muscularized (≥75% SMC actin around the vessel). The percentage of pulmonary vessels in each muscularization category was determined by dividing the number of vessels in that category by the total number counted in the same experimental group.

Elastica-van-Gieson staining according to common histopathological procedures was performed to allow the assessment of the medial wall thickness (MWT), the ratio of neointima/media and the occlusion score. Therefore, 100 pulmonary vessels with a diameter of 20–50 µm were analyzed at a 630-fold magnification. Wall thickness (WT) was defined as the mean distance between the lamina elastic externa and the vessel lumen. Medial area is the area between the lamina elastica interna and the lamina elastica externa. Altogether with the external vessel diameter (EVD), those values were used to calculate the medial wall thickness by the following formula: MWT = $(2 \times WT/EVD) \times 100$. This value is placed into the equation to determine the ratio of the neointima/media where the neointima is characterized as the cellular layer between the lamina elastic interna and the pulmonary vessel lumen.

Next, the pulmonary vessels were analyzed for occlusive lesions on hematoxylin/eosin slides and categorized into open (≥75%), partially closed (75–25%) and closed (<25%) in respect to the relative free area of their vessel lumen, which is defined as the area within the lamina elastic interna. A quantitative analysis of luminal obstruction giving rise to the occlusion score was performed by counting at least 100 small pulmonary arteries per lung section from each rat.

For PCNA detection, FFPE tissue blocks were cooked in rodent decloaker (Biocare medical by Zytomed Systems GmbH) for antigen retrieval after dehydration (by xylol and ethanol) and incubated in $H_2O_2$/methanol. After proteinase K treatment (Novocastra-RE-7160 from Leica Mikrosysteme Vertrieb GmbH, Wetzlar, Germany) and blocking with 10% BSA followed by blocking with Rodent Block R reagent (Biocare medical by Zytomed Systems GmbH), a mouse anti-PCNA antibody (Cell Signaling Technology, Danvers, MA, USA) was applied in a 1:40 dilution overnight. Signal development was carried out with the Mouse-on-Rat HRP-polymer kit (Biocare medical by Zytomed Systems GmbH) with the Nova RED peroxidase substrate (Vector laboratories, Burlingame, CA, USA) leading to a reddish cellular appearance. For those stainings, haematoxylin was used as a counterstain. Calculation of the proliferation index was performed by counting PCNA-positive pulmonary vascular cells (20–50 µm in diameter) out of hundred vessels at a 400-fold magnification. The proliferation index was defined as the number of PCNA-positive cells per pulmonary vessel given in %. All analyses were performed in a blinded fashion.

**Reporting summary**. Further information on research design is available in the Nature Research Reporting Summary linked to this article.

## Data availability

The data underlying all findings of this study are available from the corresponding author upon reasonable request and are provided as a separate source data file.

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

## Acknowledgements

We would like to thank Kerstin Gernert, Christina Vroom, Sophia Bernhardt, Ewa Bieniek, Katharina Wolthaus, Elena Jenike, Mona Höret, Uta Eule, Gabrielle Dahlem, and Susanne Ziegler for excellent technical assistance. We thank Gabriela Michel, and Holger Hackstein from the cell sorting core facility of the Institut für Immunologie und Transfusionsmedizin (IMT). We also thank Bruno Poettker and Ingrid Henneke for their help in the design and in writing of the animal experiment proposals. We thank Rory

Morty for critical proofreading of our manuscript. This work was funded by the Deutsche Forschungsgemeinschaft (DFG, German Research Foundation)—Projektnummer 268555672—SFB 1213, project A08 and CP02 and the Cardio-Pulmonary Institute (CPI), EXC 2026, Project ID: 390649896 (N.W., F.G., H.A.G., R.T.S., W.S.). Furthermore, this project has received funding from the European Research Council (ERC) under the European Union's Horizon 2020 research and innovation program (grant agreement No 771883) (D.J.).

## Author contributions

A.W., R.T.S. participated in the design, performance, and interpretation of the experiments and results. In addition, they participated in the analysis of the data and drafting and revising of the manuscript and provided administrative, technical, and supervisory support. A.W. performed portions of cellular experiments, especially cytotoxicity and apoptosis assays, and participated in the design and interpretation of the experiments and results. She analyzed the data and was responsible for drafting and revision of the manuscript. M.C.N. performed most of the cellular experiments and participated in the design, interpretation, and analysis of the data and drafting of the manuscript. D.Y. conducted peptide-based kinase activity measurements and parts of the western blotting experiments, including drafting the materials and methods section concerning this matter. He also participated in the interpretation and analysis of the data. B.K. performed echocardiographic in vivo measurements and participated in the interpretation and analysis of the data, including drafting the corresponding materials and methods section. B.C.S. performed certain cell culture and real-time PCR experiments and participated in the interpretation and analysis of the results. L.N., D.J., C.R. provided human specimens from healthy individuals and IPAH patients, including lung tissue and primary pulmonary arterial smooth muscle cells. N.B. supervised and assisted in flow cytometric measurements and analysis. P.D. performed pathological evaluation of distinct organs from Su/Hox model to detect any potential adverse events. S.P. supervised the performance of the in vivo experiments (e.g., with the Su/Hox model) at our collaborative facility in Bad Nauheim. N.W., H.A.G., F.G., W.S., R.T.S. participated in the design and interpretation of the study of portions of the experiments and results. In addition, they participated in revising the manuscript and provided administrative, technical, and supervisory support.

## Additional information

**Competing interests:** The authors declare no competing interests.

