## [Peer Review File · Nature Communications]

Reviewers' comments:

Reviewer #1 (Remarks to the Author):

The authors provide a compelling story that supports the use of CDK inhibitors as a therapeutic in PH. This is supported by in vitro studies using human pulmonary artery smooth muscle cells showing kinase activity; CDK expression; and, effects on proliferation, metabolism, and apoptosis. Corresponding in vivo studies in the mouse hypoxia and rat MCT model show that one CDK inhibitor is too toxic for use in the model while the other decreases indices of PH. A few points for consideration:

1. The studies that examine kinase activity profiling are interesting, but this isn't a true kinome as it's a limited array with no network connections shown. This should be added or be renamed. Kinase activity profiling is based on a sample size of 3 controls and 3 PAH patients. Although the authors refer to the results as homogeneous, there is significant heterogeneity in the PAH patients with much of the "signal" generated by one patient. This needs to be addressed and some information about these individuals provided to show that they are representative. It also isn't clear what CDKs are shown by the platform that led the authors to settle on the study of CDKs as the only one that is apparent from the figure is CDK7, and retinoblastoma protein. Also, Lines 117-119: comments that there is a "homogeneous signal for almost all populations within each condition" requires further explanation. Is this referring to the multiple testing runs that are done in the processing phase or response between subjects?
2. To support the concept that the CDKs, cyclins, and RB are differentially expressed in pulmonary artery smooth muscle cells in the pulmonary arteries, there should be some demonstration of this in human tissue sections of distal pulmonary arteries using immunohistochemistry techniques. Similarly, there is also no demonstration of CDKs, cyclins, and retinoblastoma protein in sections of the vessels from the rodent models. Presenting these data are a standard for the field and necessary. Since the point of the study is that the drug leads to cell cycle arrest that affects structural remodeling in PH, the reader should be able to see the structural remodeling (at level of individual distal pulmonary artery and a lower magnification of the lung to see multiple vessels).
3. The choice of the mouse hypoxia model is unusual as no studies are done using smooth muscle cell knockout mice, which should be considered to provide more information about the suitability of what CDK should be targeted. It would also answer the question of whether or not inhibiting smooth muscle cell CDKs is important for vascular remodeling or if there has to be some effect on pulmonary artery endothelial cells (see #4). Instead, studies should be done in the Sugen/Hypoxia model that more closely resembles human disease.
4. The studies provide no information on the effect of CDK inhibition on the pulmonary artery endothelium. Since you are using systemic synthetic inhibitors, it is important to evaluate this as well at a cellular level and in the vessels.

5. The effects of the CDK inhibitors either can't be determined – too toxic – or are very modest. Here, off-target effects of the inhibitors must be shown by histology of other tissues, which are also affected by PH in humans and the animal models. In certain instances, blood chemistries should also be provided. It's not enough to just show that the CDKs are expressed in other tissues in a normal animal model. There is some impetus to explain why one of the drugs, dinaciclib was not tolerated while the other drug, which has an upstream target, is better tolerated. Is this due to the model (MCT), PH in general, or effect on another organ system?

6. For Figure 1B, please provide a densitometry of p-ERK/ERK graph. Although there is more p-ERK in PAH cells, there is also more total ERK. Also, why aren't there two bands in the control samples? Was the same antibody used?

7. The in vivo studies report very limited hemodynamic (systemic and right heart) data and the echocardiograms present very limited data. Full systemic and right heart hemodynamics needs to be added (i.e., heart rate, arterial blood pressure, SVR, etc.) and standard information such as weights, including pre-model, time of introduction of CDK inhibitor, and end of study.

Reviewer #2 (Remarks to the Author):

In this manuscript the Authors focused on pulmonary arterial hypertension (PAH). By performing a kinome profiling of human pulmonary arterial smooth muscle cells (PAMSC) derived from healthy individuals and patients affected by idiopathic PAH (IPAH), the Authors identified cyclin-dependent kinases (CDKs) as being overactivated in IPAH-PAMSC. They then investigated the effect of the treatment of PAMSC with the CDK inhibitors dinaciclib and palbociclib, on the cell cycle. Finally the Authors investigated the potential therapeutic efficacy of the CDK4 and CDK6 inhibitor palbociclib in a PAH rat model. The topic is interesting, however I have several concerns as following described.

Major points

Figure 1.

-Figure 1b. The Authors state that in order to demonstrate a different phenotype of human pulmonary arterial smooth muscle cells (HPASMC) derived from individuals with idiopathic pulmonary arterial hypertension (IPAH-HPASMC) versus HPASMC derived from healthy individuals, they investigated the activation of ERK1/2 signaling pathway. It is not clear why the ERK1/2 signaling pathway was selected, the Authors should explain better the rationale of this choice. In addition, assuming that the numbers "#1", "#2" and "#3" refer to individual patients, this aspect should be clarified in the figure legend.

-The Authors affirm that HPASMC derived from IPAH patients are hyperproliferating cells, however they do not show any characterization of these cells in terms of proliferation potential in comparison with HPASMC derived from healthy individuals.

-Figure 1c-e. The Authors state that the CDK activity is up-regulated in IPAH patients-derived HPASMC (fig. 1c) when compared to HPASMC derived from healthy individuals. However, the data relative to the activity of CDKs are not clearly shown. The resolution of the figures 1d and 1e is low and the text in the figures is not readable.

Figure 2.

-The Authors investigated the expression of CDK2, CDK4 and CDK6 in human tissues derived from healthy and IPAH patients. However, the Authors do not clearly explain the reason why they focused on these three CDKs.

Figure 3.

-The expression of CDK2, CDK4 and CDK6 on mRNA level (figure 3d) and protein level (figure 3e) is not consistent. For example, the brain and muscle tissues showed the highest mRNA levels of CDK4 when compared to the other organs, however CDK4 was undetectable on protein level in both brain and muscle. The Authors should comment on this aspect.

Figure 4 and figure 5.

-In figure 4 and in figure 5a-c the Authors show the effect of the treatment with dinaciclib and palbociclib on cell proliferation, DNA synthesis etc, however it is not clear whether these experiments were performed by using HPASMC derived from healthy or IPAH patients. The Authors should clarify this aspect. Also, the Authors did not show any data relative to the measure of the IC50, for both the drugs, in their system.

-In addition, it is not clear why the Authors used as control the cells treated with basal media (BM) and not with GM-2 media as the cells treated with dinaciclib and palbociclib.

-In figure 5a and 5b the Authors show the effect of the treatment of the cells with dinaciclib and palbociclib on the cell cycle and cell apoptosis. In lines 195-197, the Authors state that there was no sign of cell death induction upon treatment of the cells with the drugs, since the fraction of cells in SubG1 phase did not increase (figure 5a-b). However, in lines 200-204 the Authors affirmed that dinaciclib triggered apoptosis-mediated cell death (figure c). The Authors should explain this discrepancy. It is opinion of this reviewer that a deeper characterization of the apoptosis process,

also under a molecular point of view (i.e. by investigation the potential involvement of caspases), would be necessary.

-In figure 5d the Authors show the effect of the treatment of healthy HPASMCs on the levels of phosphorylated and total Rb. Here, both dinaciclib and palbociclib reduced the levels of phosphorylated Rb (pRb), without affecting the levels of total Rb. By contrast, in the western blot shown in figure 5e, the levels of both pRb and total Rb were reduced in healthy HPASMCs treated with both dinaciclib and palbociclib. The Authors should explain this discrepancy. Also, concerning the levels of p21, the Authors do not comment on the reported differences between cells from healthy and pathological individuals or among cells treated with the different drugs. Finally, in figure 5d, the Authors show the detection of PCNA, which is not mentioned neither in the manuscript nor in the figure legend.

-In the experiments shown in figure 5e the Authors introduced the use of imatinib, however they do not explain the rationale of using this drug and of comparing it to dinaciclib and palbociclib.

-It is opinion of this reviewer that the levels of phosphorylation of CDK2, CDK4, CDK6 and CDK9 and their total levels should be included in the western blot analysis. Indeed, since these CDKs are the targets of the used drugs, their characterization is crucial.

Figure 7

-In figure 7i and 7j, the Authors show that the treatment of MCT-treated rats with palbociclib led to a reduction in the mRNA levels of CCN2A and CDK1. From these data the Authors concluded (lines 535-534) that "cellular proliferation was almost completely blocked in the animals treated with palbociclib, as evidenced by the reduced levels of two important transcriptional CDK-Rb-E2F downstream target genes, CCN2A and CDK1". It is opinion of this reviewer that the observed reduction in the mRNA levels of CCN2A and CDK1 is not enough to conclude that cell proliferation was blocked. Additional experiments, such as the immunohistochemical detection of proliferation markers like ki67 should be performed. Also, an evidence that palbociclib actually affected the activity of the targeted CDK4 and CDK6 in vivo should be included. The levels of phosphorylated and total CDK4 and CDK6 should be determined (for example by immunohistochemistry).

Minor points

-Line 219, please correct "transcriptional target genes of activated by E2F" with "transcriptional target genes of E2F".

-In the Results section, lines 98-107, the Authors explained in a really detailed manner the experimental procedure followed for the acquisition of the data shown in figure 1c-1e. Maybe this part can be moved to the Material and Method section.

-Figure 2, line 767: Please correct "(b-n)" with "(h-n)".

-Figure 5: Please make the text in figure 5a bigger.

Point-by-point response to the comments of reviewer #1

Q1. The studies that examine kinase activity profiling are interesting, but this isn't a true kinome as it's a limited array with no network connections shown. This should be added or be renamed. Kinase activity profiling is based on a sample size of 3 controls and 3 PAH patients. Although the authors refer to the results as homogeneous, there is significant heterogeneity in the PAH patients with much of the "signal" generated by one patient. This needs to be addressed and some information about these individuals provided to show that they are representative. It also isn't clear what CDKs are shown by the platform that led the authors to settle on the study of CDKs as the only one that is apparent from the figure is CDK7, and retinoblastoma protein. Also, Lines 117-119: comments that there is a "homogeneous signal for almost all populations within each condition" requires further explanation. Is this referring to the multiple testing runs that are done in the processing phase or response between subjects?

R1: Thank you for this comment. The reviewer is indeed right; this approach is not directly measuring the activity of all individual kinases, which build up the complete kinome. There is a variety of different techniques (e.g., *KinomeView Profiling* platform, *KINOMEscan* platform, phospho-proteome analysis by mass spectrometry) allowing for the analysis of the kinome with each of them harboring crucial limitations. The peptide-based system used in the present study uses only a distinct set of peptides which have carefully been selected to cover the substrate variety of all kinases of the whole kinome. The phosphorylation status of those 144 peptides presented on the chip allow for the prediction of corresponding kinases (figure 1c; revised) that are upstream of those substrates.

Figure 1. (c) Bar chart displaying the overall prediction of kinases with higher activity in IPAH-patient derived samples. Normalized kinase statistics is a mathematical based algorithm indicating the estimated relative kinase activity while the specificity score reflects the reliability and accuracy of the prediction.

This kind of investigation does not allow for a comprehensive signaling network presentation which could only be done on the level of the whole phospho-proteome. Detailed explanation of the technique has been discussed in the revised version of the manuscript in the supplement section (page 56-57, line 1167-1193). Next, we addressed the point of heterogeneity within the samples of each group e.g. healthy and IPAH, by physically performing another data assessment, increasing the sample size to n=6 with subsequent analysis of the data. The individual data sets are included in the supplementary material (figure S1; revised).

Figure S1. Heat map from the peptide micro array obtained from all individual samples. Raw data for all individual HPASMC samples from both entities presented as a heat map with log-transformed fluorescence signals due to kinase-mediated substrate serine/threonine phosphorylation. Based on those signal intensities an upstream kinase activity was predicted with increased levels for distinct CDKs.

Despite a variation between the different individual healthy HPASMCs, as well as in the IPAH-patient derived cells, increased CDK activity is a common feature of all IPAH-HPASMCs. In the revised version, we decided to present a heat map showing the mean phosphorylation pattern of multiple samples for the two groups (figure 1a, b; revised).

Figure 1. (a) Mean value of raw data for all individual samples, e.g. HPASMCs from healthy individuals (n=5) and IPAH patients (n=6), presented as a heat map of log-transformed fluorescence signals upon substrate serine/threonine phosphorylation that are used to perform an upstream kinase analysis. (b) Mean value of a two-group comparison for all samples showing the global changes in the serine/threonine phosphorylation pattern between both HPASMCs entities under basal media conditions.

We now also added the detailed results from the computational upstream kinase prediction with further kinases of potential interest in the form of a bar chart (figure 1d; revised).

Figure 1. (d) Two-group comparison for the four selected CDKs only including the phosphorylation pattern of their particular peptide substrate sets on the peptide array.

Based on the above-mentioned analysis, we focused on CDKs as these data sets clearly demonstrates CDK6, CDK9, CDK4 and CDK2 with a highest prediction value in IPAH-derived HPASMCs (figure 1c; revised). For better clarity, we (i) changed the format of the two group comparison description, (ii) included the computational upstream kinase prediction in the revised version of the manuscript, and (iii) provided a phosphorylation profile for upstream kinase prediction of CDK2, CDK4, CDK6, and CDK9 (figure 1d; revised). Although the signals are not homogenous and display distinct patterns of phosphorylation, we still identified increased CDK activity as a common feature for all IPAH-HPASMCs. By this, our former statement in lines 117-119 has been clarified.

Q2. To support the concept that the CDKs, cyclins, and RB are differentially expressed in pulmonary artery smooth muscle cells in the pulmonary arteries, there should be some demonstration of this in human tissue sections of distal pulmonary arteries using immunohistochemistry techniques. Similarly, there is also no demonstration of CDKs, cyclins, and retinoblastoma protein in sections of the vessels from the rodent models. Presenting these data are a standard for the field and necessary. Since the point of the study is that the drug leads to cell cycle arrest that affects structural remodeling in PH, the reader should be able to see the structural remodeling (at level of individual distal pulmonary artery and a lower magnification of the lung to see multiple vessels).

R2: We would like to thank the reviewer for this valuable suggestion. As suggested, we performed immunohistochemical staining of CDK2, CDK4, CDK6, P-Rb, PCNA along with alpha-SMA (a marker specific for smooth muscle cells) and vWF (a marker specific for endothelial cells). Immunohistochemical staining of human donor and IPAH-lungs showed that CDK2/4/6 expression was increased in the medial layer and the obliterated lumen of IPAH pulmonary arteries. Increased CDK2/4/6 expression correlated well with increased PCNA expression in IPAH remodelled vessels (figure 2a-c; revised), suggesting increased cell proliferation of smooth muscle cells *in vivo* in human PAH lungs.

Figure 2: Immunohistological staining of distinct CDKs and P-Rb activation in human lung specimen. Representative images of serial sections from healthy (n=5) and IPAH-patient derived lungs (n=5) analyzed for CDK2 (a), CDK4 (b), and CDK6 (c), with corresponding IHC for P-Rb protein and its downstream target gene and common proliferation marker PCNA (proliferation cell nuclear antigen). Cellular identity was visualized by antibodies against α-SMA (alpha-smooth muscle actin) and vWF (von-Willebrand factor). Images were taken at 400-fold magnification. Scale bar 20 μm.

Results referring to this question can be found in the revised version of the manuscript (page 7-8, line 157-165). Furthermore, to assess the influence of palbociclib on structural remodeling in experimental models of PH, we performed double staining with anti- α -actin and anti-vWF, followed by quantification of the degree of muscularization of small peripheral pulmonary arteries. Notably, palbociclib treatment in two experimental models of PH (MCT and Su/Hox) markedly reduced the number of fully muscularized arteries (figure 6a, 6b, 8a, 8b; revised) along with reduced number of PCNA positive cells (figure 6c and 8d; revised), confirming that palbociclib induces cell cycle arrest and subsequently improves structural vascular remodeling.

Figure 6: Ex vivo analyses of lung tissue for reversal of remodeling and in vivo drug efficacy in the MCT rat model. (a) The degree of muscularization of small pulmonary arteries (diameter 25 - 50 μ m) was determined ex vivo via immunohistological staining of lung sections for vWF (brown) and α -SMA (violet) together with methylgreen for counterstaining. M: fully muscularized; P: partially muscularized; N: non-muscularized. Representative images for all three study groups are shown. (b) Medial wall thickness of vessels with a diameter of 20-50 μ m was determined by Elastica-van-Gieson staining and is presented as percentage. Data are presented as mean \pm SEM and statistical analysis was performed using one-way ANOVA with Newman-Keuls post-hoc test for multiple comparisons; ** $p < 0.01$ for MCT+Palbo versus MCT; \S $p < 0.05$, $\S\S$ $p < 0.01$ for MCT+Palbo versus healthy; $\#\#\#$ $p < 0.001$ for healthy versus MCT.

Figure 8: Ex vivo analyses of lung tissue for reversal of remodeling and in vivo drug efficacy in the Su/Hox model. (a) The degree of muscularization of small pulmonary arteries (diameter 25 - 50 µm) was determined ex vivo via immunohistological staining of lung sections for vWF (brown) and α-SMA (violet) together with methylgreen for counterstaining. M: fully muscularized; P: partially muscularized; N: non-muscularized. Representative images for all three study groups (Nox, Su/Hox, Su/Hox+Palbo) are shown. (b) The medial wall thickness as well as the corresponding ratio of neointima/media of the depicted study groups were determined by Elastica-van-Gieson staining. Data are presented as mean±SEM and statistical analysis was performed using one-way ANOVA with Newman-Keuls post-hoc test for multiple comparisons; ** p < 0.01, *** p < 0.001 for MCT+Palbo versus MCT; § p < 0.05, §§ p < 0.01, §§§ p < 0.001 for MCT+Palbo versus healthy; ### p < 0.001 for healthy versus MCT.

Figure 6. (c) Proliferation index as a quantitative measure of PCNA-positive cells (purple) per vessel. Representative images (using hematoxylin/eosin as counterstain) for all three study groups are included. Data are presented as mean±SEM and statistical analysis was performed using one-way ANOVA with Newman-Keuls post-hoc test for multiple comparisons; *** p < 0.001 for MCT+Palbo versus MCT; ### p < 0.001 for healthy versus MCT.

Figure 8. (d) Proliferation index as a quantitative parameter of PCNA-positive cells (purple) per vessel. Representative images (using hematoxylin/eosin as counterstain) for all three study groups are included. Data are presented as mean±SEM and statistical analysis was performed using one-way ANOVA with Newman-Keuls post-hoc test for multiple comparisons; * $p < 0.05$ for Su/Hox+Palbo versus Su/Hox.

As requested by the reviewer, vascular remodeling changes both at the level of individual distal pulmonary artery and a lower magnification of the lung to see multiple vessels is provided (figure S7; revised).

Figure S7: Structural remodeling at the level of individual pulmonary arteries for both experimental PAH models. Representative images from microscopic fields of interest with multiple vessels for lungs explanted from the MCT (**a-c**) and the Su/Hox (**d-f**) rat model stained for α -SMA and vWF. Magnifications from left to right: 25-fold, 50-fold, 100-fold, and 200-fold.

Results referring to this question can be found in the revised version of the manuscript (page 11, line 263-268; page 12, line 292-298).

Q3. The choice of the mouse hypoxia model is unusual as no studies are done using smooth muscle cell knockout mice, which should be considered to provide more information about the suitability of what CDK should be targeted. It would also answer the question of whether or not inhibiting smooth muscle cell CDKs is important for vascular remodeling or if there has to be some effect on pulmonary artery endothelial cells (see #4). Instead, studies should be done in the Sugen/Hypoxia model that more closely resembles human disease.

R3: As suggested by the reviewer, we performed additional experiments addressing therapeutic efficacy of palbociclib in the Su/Hox rat model of PH that closely resembles human disease. Similar to the MCT experiments, palbociclib was given to SU5416- and hypoxia-treated rats (Su/Hox) after the establishment of the disease. A scheme of the treatment groups is provided in revised figure 7a according to previous publications of our group^{1,2}. Notably, we found that palbociclib treatment significantly reduced right ventricular systolic pressure (RVSP), pulmonary vascular resistance (PVR) and improved cardiac performance. Additional parameters, which have been measured by invasive (right heart catheterization) and non-invasive (echocardiography) techniques are also given in figure 7 and table 1b in the revised manuscript. Results referring to this question can be found in the revised version of the manuscript (page 12, line 277-285).

Figure 7: Effects of palbociclib on disease progression in the Su/Hox rat model of pulmonary arterial hypertension. (a) Scheme of animal treatments. Echocardiograms were recorded at baseline, 21 days after Su5416 administration at the beginning of hypoxia exposure and again on day 35 after a 14-day period of treatment with 75 mg/kg body weight palbociclib (or placebo) daily by gavage under re-exposure to normoxic conditions. (b-l) On day 35, hemodynamics and cardiac function were assessed in vivo 14 days after treatment with with palbociclib (Su/Hox+Palbo, n=7) (under hypoxia exposure). Similarly, normoxic rats

(n=8) and animals injected with Su5416 and placebo (under hypoxia exposure) (Su/Hox, n=8) were used as control groups. Data are presented as mean±SEM and statistical analysis was performed using one-way ANOVA with Newman-Keuls post-hoc test for multiple comparisons; * p < 0.05, ** p < 0.01, *** p < 0.001 for Su/Hox+Palbo versus Su/Hox; § < 0.05, §§ p < 0.01, §§§ p < 0.001 for Su/Hox+Palbo versus Nox; # < 0.05, ## p < 0.01, ### p < 0.001, #### p < 0.0001 for Nox versus Su/Hox.

a

Parameter	MCT; n=5		MCT + Palbociclib; n=6	
	Pre-treatment	Post-treatment	Pre-treatment	Post-treatment
BW [g]	452 ± 30	451 ± 37	420 ± 37	405 ± 33
HR [bpm]	342 ± 34	339 ± 40	342 ± 39	339 ± 24
SVI [μ l 100g BW ⁻¹]	46.8 ± 4.5	34.7 ± 9.9 §	49.7 ± 7.7	55.4 ± 10.7 ##
CI [ml min ⁻¹ per 100g BW]	15.9 ± 1.9	11.9 ± 3.9 §	16.7 ± 1.5	17.7 ± 2.4 #
RVID [mm]	4.2 ± 0.2	6.0 ± 0.8 §§§§	4.1 ± 0.3	4.5 ± 0.4 ###
TAPSE [mm]	2.2 ± 0.2	1.8 ± 0.3 §	2.4 ± 0.1	2.3 ± 0.2 ##

Table 1: Further hemodynamic description of both experimental PAH models. Data are presented as mean±SD and statistical analysis was performed using one-way ANOVA with Newman-Keuls post-hoc test for multiple comparisons (a) MCT rat model; § p < 0.05, §§§§ p < 0.0001 for post- versus pre-treatment of MCT; # p < 0.05, ## p < 0.01, ### p < 0.001 for post-treatment MCT+Palbociclib versus MCT.

Further assessment of structural remodeling suggests that palbociclib not only reduces muscularization of distal pulmonary arteries, but also neointima formation, arguing its influence on both hyper-proliferative smooth muscle cells and endothelial cells (Figure 3, S5; revised). Results referring to this question can be found in the revised version of the manuscript (page 12, line 290-292).

Figure 3. Cells were synchronized in basal media (BM) and then treated with the indicated concentrations of palbociclib or DMSO as control in the presence of growth media (GM-2) for 24 h. IC₅₀ graphs determining the half maximal inhibitory concentration of palbociclib to block GM-2 induced proliferation of healthy HPASMCs (b) and of IPAH-HPASMCs (d) as measured by BrdU incorporation (relative absorbance [A_{370nm}-A_{492nm}]). IC₅₀ values were calculated from measurements of two individual primary cell isolates (run twice in triplicates each) by the non-linear regression (curve fit) module with a variable slope using four parameters as provided by GraphPad Prism software.

Figure S5. (b) Healthy HPAECs were synchronized in endothelial specific basal media (ECBM supplemented with 0.5% FCS) and then treated with the indicated concentrations of palbociclib or DMSO as control in the presence of growth media (ECGM) for 24 h. IC₅₀ graphs determining the half maximal inhibitory concentration of palbociclib (**b**) to block ECGM induced proliferation of healthy HPAECs as determined by BrdU incorporation (relative absorbance [A_{370nm}-A_{492nm}], n=1). IC₅₀ values were calculated from measurements of one primary cell isolate (run in two independently performed triplicates) by the non-linear regression (curve fit) module with a variable slope using four parameters as provided by GraphPad Prism software.

Thus, this set of additional animal experiments clearly provides further evidence of the therapeutic potential of palbociclib in pulmonary hypertension.

Q4. The studies provide no information on the effect of CDK inhibition on the pulmonary artery endothelium. Since you are using systemic synthetic inhibitors, it is important to evaluate this as well at a cellular level and in the vessels.

R4: As requested by the reviewer, we performed additional series of experiments to study the expression of CDKs as well as the effect of CDK inhibition on human pulmonary artery endothelial cells. As shown in figure 2 (revised), a co-localization of CDKs and P-Rb with vWF was observed in pulmonary vessels, suggesting that CDKs and P-Rb were also increased in endothelial cells contained in the obliterated lumen of the small remodeled pulmonary vessels. Furthermore, dinaciclib and palbociclib, both similarly decreased the proliferation and cellular CDK-Rb-E2F-signaling in primary human pulmonary arterial endothelial cells *in vitro* (figure S5; revised) and neointima formation in Su/Hox rat model (figure 8b, revised).

Figure S5: Evaluation of IC_{50} concentrations of the CDK inhibitors dinaciclib and palbociclib on proliferation, and their effects on CDK-Rb-E2F signaling in healthy HPAECs. Cells were synchronized in endothelial specific basal media (ECBM supplemented with 0.5% FCS) and then treated with the indicated concentrations of dinaciclib or palbociclib or DMSO as control in the presence of growth media (ECGM) for 24 h. IC_{50} graphs determining the half maximal inhibitory concentration of dinaciclib (**a**) and palbociclib (**b**) to block EC-GM induced proliferation of healthy HPAECs as determined by BrdU incorporation (relative absorbance [$A_{370nm} - A_{492nm}$], $n=1$). IC_{50} values were calculated from measurements of one primary cell isolate (run in two independently performed triplicates) by the non-linear

regression (curve fit) module with a variable slope using four parameters as provided by GraphPad Prism software. Representative microscopic images from distinct cell culture conditions (c) were taken with a 400-fold magnification prior the performance of representative Western blots (d) for the detailed analyses of cellular signaling on protein level after 24 hours of treatment with dinaciclib, palbociclib, or staurosporine (SSP) in healthy HPAECs with regard to CDK activation and downstream Rb-E2F pathway induction. CCNA2 (left) and CDK1 (right) mRNA expression (normalized to GAPDH as a house keeping gene) of healthy HPAECs (e) upon 24 hours of inhibitor exposure. All data from one primary cell isolates (run twice in triplicates each) are presented as mean±SEM of the n-fold change ($2^{-\Delta\Delta C_t}$) compared with DMSO treated control samples (black bar). Statistical analysis was performed using one-way ANOVA with Newman-Keuls post-hoc test for multiple comparisons; *** p < 0.001. P-values for distinct conditions are only given for their comparison with DMSO-treated control cells (black bar), and the significance of the difference between the three conditions with various concentrations of each of the CDK inhibitors is represented as follows: § p < 0.05, §§ p < 0.01, §§§ p < 0.001.

Figure 8. (b) The medial wall thickness as well as the corresponding ratio of neointima/media of the depicted study groups were determined by Elastica-van-Gieson staining allowing the calculation of the vessel occlusion score. Data are presented as mean±SEM and statistical analysis was performed using one-way ANOVA with Newman-Keuls post-hoc test for multiple comparisons; ** p < 0.01, *** p < 0.001 for Su/Hox+Palbo versus Su/Hox; §§§ p < 0.001 for Su/Hox+Palbo versus Nox; ### p < 0.001 for Nox versus Su/Hox.

These results are now included in the revised version of the manuscript on page 9-10, line 219-229 and page 12, line 290-292).

Q5. The effects of the CDK inhibitors either can't be determined – too toxic – or are very modest. Here, off-target effects of the inhibitors must be shown by histology of other tissues, which are also affected by PH in humans and the animal models. In certain instances, blood chemistries should also be provided. It's not enough to just show that the CDKs are expressed in other tissues in a normal animal model. There is some impetus to explain why one of the drugs, dinaciclib was not tolerated while the other drug, which has an upstream target, is better tolerated. Is this due to the model (MCT), PH in general, or effect on another organ system?

R5: We agree with the reviewer that models of PH can contribute to the toxic effects and no animal model per se completely recapitulates human PAH vascular pathologies. Particularly, monocrotaline via reactive metabolites (e.g. dehydromonocrotaline) was shown to influence several organs apart from pulmonary vasculature and can induce myocarditis, liver injury, kidney injury and even muscle impairment³⁻⁵. Although we strongly believe that the MCT model of PH is a valuable model to assess the therapeutic effects of anti-remodeling compounds on the pulmonary vasculature, the toxic effects of MCT on other organs questions the interpretation of toxic /off-target effects of any compound in this model. In contrast, Su5416 in combination with hypoxia (Su/Hox) predominantly influences the pulmonary vasculature, making it as a suitable model to assess also the potential toxicity of the drugs. Therefore, we thank the reviewer for suggesting us to address the therapeutic efficacy of palbociclib in the Su/Hox rat model. Importantly, we found that palbociclib treatment significantly reduced RVSP, PVRI and improved cardiac performance (see R3). A pathologist blinded to the experimental groups, evaluated histological or structural abnormalities in Su/Hox+palbociclib or placebo treated rat organs (left ventricle, liver, kidney and intestine) and could not find evidence for potential toxicity of palbociclib (figure S9; revised).

Figure S9. (b) Representative histopathological images of different organs of H&E stained FFPE sections from the Su/Hox rat model.

In addition, we performed a complete analysis of blood cells in the treated rats. In accordance with the known side-effects profile of palbociclib available from public-available data bases (https://www.accessdata.fda.gov/drugsatfda_docs/label/2016/207103s002lbl.pdf), we observed a decrease in white blood cells, lymphocytes neutrophils, and platelets (Figure S8, revised).

Figure S8: Blood analysis of laboratory rats. Blood analysis from samples derived from all rats of the Su/Hox study. Distinct parameters indicating any possible negative side effects of palbociclib based on the known toxicology profile were analyzed by the flow cytometry. * $p < 0.05$; ** $p < 0.01$, *** $p < 0.001$ for Su/Hox+Palbo versus Su/Hox; § $p < 0.05$; §§§ $p < 0.001$ for Su/Hox+Palbo versus Nox; ### $p < 0.01$, #### $p < 0.001$ for Nox versus Su/Hox.

This data suggest that the side effect profile of palbociclib in the context of pulmonary hypertension is comparable to the primary indication. Results referring to this question can be found in the revised version of the manuscript (page 13, line 306-323).

Q6. For Figure 1B, please provide a densitometry of p-ERK/ERK graph. Although there is more p-ERK in PAH cells, there is also more total ERK. Also, why aren't there two bands in the control samples? Was the same antibody used?

R6: During the revision process, several graphs have been rearranged and additional experiments were included. We decided to remove this particular Western blot of p-ERK/ERK, as we did not discuss the effects of stimulation of HPASMCs with a multiple-growth factor (MGF) cocktail. Furthermore, we focused on the investigation of cell cycle regulatory proteins (e.g. CDKs, Rb, cyclins) to streamline the experiments.

Q7. The in vivo studies report very limited hemodynamic (systemic and right heart) data and the echocardiograms present very limited data. Full systemic and right heart hemodynamics needs to be added (i.e., heart rate, arterial blood pressure, SVR, etc.) and standard information such as weights, including pre-model, time of introduction of CDK inhibitor, and end of study.

R7: As suggested by the reviewer, we provided now all hemodynamic (e.g. cardiac index, pressures, resistance), echocardiographic (e.g. RVID, TAPSE, stroke volume) and standard information such as body weight, heart rate pre- and post- treatment in both animal models of PH (figure 5, 7, and table 1; revised). The treatment protocols are now presented as schemes in the revised version of the manuscript (figure 5a, 7a; revised).

Figure 5: Effects of palbociclib on disease progression in the MCT rat model of pulmonary arterial hypertension. (a) Scheme of animal treatments. Echocardiograms were recorded at baseline, 21 days after MCT administration and again on day 35 after a 14-day period of treatment with 75 mg/kg body weight palbociclib (or placebo) daily by gavage. (b-l) On day 35, hemodynamics and cardiac function were assessed *in vivo* 14 days after treatment with palbociclib (MCT+Palbo, n=5). Similarly, healthy rats (Healthy, n=6) and animals injected with MCT and placebo (MCT, n=5) were used as control groups. Data are

presented as mean±SEM and statistical analysis was performed using one-way ANOVA with Newman-Keuls post-hoc test for multiple comparisons; * p < 0.05, ** p < 0.01, *** p < 0.001 for MCT+Palbo versus MCT; § < 0.05, §§ p < 0.01, §§§ p < 0.001, §§§§ p < 0.0001 for MCT+Palbo versus healthy; # < 0.05, ## p < 0.01, ### p < 0.001, #### p < 0.0001 for healthy versus MCT.

Figure 7: Effects of palbociclib on disease progression in the Su/Hox rat model of pulmonary arterial hypertension. (a) Scheme of animal treatments. Echocardiograms were

recorded at baseline, 21 days after Su5416 administration at the beginning of hypoxia exposure and again on day 35 after a 14-day period of treatment with 75 mg/kg body weight palbociclib (or placebo) daily by gavage under re-exposure to normoxic conditions. **(b-l)** On day 35, hemodynamics and cardiac function were assessed *in vivo* 14 days after treatment with with palbociclib (Su/Hox+Palbo, n=7) (under hypoxia exposure). Similarly, normoxic rats (n=8) and animals injected with Su5416 and placebo (under hypoxia exposure) (Su/Hox, n=8) were used as control groups. Data are presented as mean±SEM and statistical analysis was performed using one-way ANOVA with Newman-Keuls post-hoc test for multiple comparisons; * p < 0.05, ** p < 0.01, *** p < 0.001 for Su/Hox+Palbo versus Su/Hox; § p < 0.05, §§ p < 0.01, §§§ p < 0.001 for Su/Hox+Palbo versus Nox; # p < 0.05, ## p < 0.01, ### p < 0.001, #### p < 0.0001 for Nox versus Su/Hox.

a

Parameter	MCT; n=5		MCT + Palbociclib; n=6	
	Pre-treatment	Post-treatment	Pre-treatment	Post-treatment
BW [g]	452 ± 30	451 ± 37	420 ± 37	405 ± 33
HR [bpm]	342 ± 34	339 ± 40	342 ± 39	339 ± 24
SVI [μ l 100g BW ⁻¹]	46.8 ± 4.5	34.7 ± 9.9 §	49.7 ± 7.7	55.4 ± 10.7 ##
CI [ml min ⁻¹ per 100g BW]	15.9 ± 1.9	11.9 ± 3.9 §	16.7 ± 1.5	17.7 ± 2.4 #
RVID [mm]	4.2 ± 0.2	6.0 ± 0.8 §§§§	4.1 ± 0.3	4.5 ± 0.4 ###
TAPSE [mm]	2.2 ± 0.2	1.8 ± 0.3 §	2.4 ± 0.1	2.3 ± 0.2 ##

b

Parameter	Su/Hox; n=8		Su/Hox + Palbociclib; n=7	
	Pre-treatment	Post-treatment	Pre-treatment	Post-treatment
BW [g]	365 ± 20	391 ± 19	363 ± 16	362 ± 28
HR [bpm]	349 ± 28	337 ± 17	368 ± 16	317 ± 42 **
SVI [μ l 100g BW ⁻¹]	35.5 ± 4.6	33.3 ± 4.9	35.3 ± 3.2	54.9 ± 11.9 **** #####
CI [ml min ⁻¹ per 100g BW]	12.4 ± 1.5	11.2 ± 1.7	12.8 ± 1.1	16.6 ± 3.5 ** ###
RVID [mm]	4.7 ± 0.3	5.3 ± 0.5 §§	4.7 ± 0.3	4.3 ± 0.5 ###
TAPSE [mm]	1.49 ± 0.07	1.3 ± 0.1 §§	1.46 ± 0.08	2.1 ± 0.1 **** #####

Table 1: Further hemodynamic description of both experimental PAH models. Data are presented as mean±SD and statistical analysis was performed using one-way ANOVA with Newman-Keuls post-hoc test for multiple comparisons **(a)** MCT rat model; § p < 0.05, §§§§ p < 0.0001 for post- versus pre-treatment of MCT; # p < 0.05, ## p < 0.01, ### p < 0.001 for post-treatment MCT+Palbociclib versus MCT. **(b)** Su/Hox model; §§ p < 0.01 for post- versus pre-treatment of Su/Hox; ** p < 0.01, **** p < 0.0001 for post- versus pre-treatment of Su/Hox+Palbociclib; ### p < 0.001, ##### p < 0.0001 for post-treatment Su/Hox+Palbociclib versus Su/Hox. BW: body weight; HR: heart rate; SVI: stroke volume index; CI: cardiac index; RVID: right ventricular internal diameter; TAPSE: tricuspid annular plane systolic excursion.

Results referring to this question can be found in the revised version of the manuscript (page 11, line 253-263; page 12, line 280-285).

Point-by-point response to the comments of reviewer #2

Figure 1.

-Figure 1b. The Authors state that in order to demonstrate a different phenotype of human pulmonary arterial smooth muscle cells (HPASMC) derived from individuals with idiopathic pulmonary arterial hypertension (IPAH-HPASMC) versus HPASMC derived from healthy individuals, they investigated the activation of ERK1/2 signaling pathway. It is not clear why the ERK1/2 signaling pathway was selected, the Authors should explain better the rationale of this choice. In addition, assuming that the numbers "#1", "#2" and "#3" refer to individual patients, this aspect should be clarified in the figure legend.

R: The activation of ERK1/2 signaling pathway was assessed to demonstrate the hyperproliferative response of healthy and IPAH-HPASMCs upon multiple growth factors (MGF) stimulation. However, as this MGF stimulation may lead to overactivation of kinases and their respective substrates, we decided to study the kinome profile under basal conditions of both donor and IPAH-HPASMCs, and thus omitted the results with MGF stimulation shown in figure 1b. We increased the sample size to n=6 with subsequent analysis of the data. The individual data sets are included in the supplementary material (figure S1; revised).

Figure S1. Heat map from the peptide micro array obtained from all individual samples. Raw data for all individual HPASMC samples from both entities presented as a heat map with log-transformed fluorescence signals due to kinase-mediated substrate serine/threonine phosphorylation. Based on those signal intensities an upstream kinase activity was predicted with increased levels for distinct CDKs.

-The Authors affirm that HPASMC derived from IPAH patients are hyper-proliferating cells, however they do not show any characterization of these cells in terms of proliferation potential in comparison with HPASMC derived from healthy individuals.

R: Previous publications from our group and others have shown that HPASMCs isolated from IPAH-patients have a higher proliferative potential ^{1,2}. Since growth factor stimulation will lead to activation of kinases and their respective substrates, we decided to study the kinome profile under basal conditions of both healthy and IPAH-HPASMCs. Under basal conditions, we have observed an increased cyclin D1, (P-)CDK2, CDK4, and (P-)CDK6, suggesting an increased proliferative capacity of IPAH-HPASMCs (figure 1e; revised).

Figure 1. (e) Western blot analysis for CDK2, CDK4, CDK6, and CDK9 activation and subsequent Rb-E2F downstream signaling in HPASMCs obtained from healthy donors and IPAH patients exposed to basal media.

In this context, we would like to draw the attention to the quantitative real-time PCR analysis of two CDK-Rb-E2F downstream target genes namely CDK1 (figure 1f; revised) and Cyclin A2 (figure S2e; revised), both of which show a significant increase in their mRNA expression in IPAH-patient derived HPASMCs.

Figure 1. (f) CDK1 mRNA expression normalized to GAPDH as reference gene in HPASMCs of healthy individuals (n=5) and IPAH patients (n=5). After starvation, cells were cultured in basal media for 24 h. All data are presented as mean±SEM of the n-fold change ($2^{-\Delta\Delta C_t}$) compared with a healthy control and analyzed statistically using a Mann-Whitney test; ** p < 0.01.

Figure S2. (e) mRNA analysis normalized to GAPDH as reference gene for CCNA2 in HPASMCs from healthy individuals and IPAH-patients (n=3, each). All data are presented as mean±SEM of the n-fold change ($2^{-\Delta\Delta C_t}$) compared with a healthy control and analyzed statistically using a Mann-Whitney test; **** p < 0.0001.

This clearly shows an up-regulation of key proteins involved in the regulation of the cell cycle and proliferation further supporting our hypothesis of CDK inhibition as a therapeutic option for PAH. Results referring to this question can be found in the revised version of the manuscript (page 6-7, line 134-149).

-Figure 1c-e. The Authors state that the CDK activity is up-regulated in IPAH patients-derived HPASMC (fig. 1c) when compared to HPASMC derived from healthy individuals. However, the data relative to the activity of CDKs are not clearly shown. The resolution of the figures 1d and 1e is low and the text in the figures is not readable.

R: In order to address the increased CDK activity in IPAH-HPASMCs, we decided to present a heat map showing the mean phosphorylation pattern of multiple samples for the two groups (figure 1a, b; revised) in the revised version of the manuscript. The n-number has been increased to n=6 and all individual data sets are included in the supplementary material (figure S1; revised). An increased CDK activity is a common feature of all IPAH-HPASMCs. We now also added the detailed results from the computational upstream kinase prediction with further kinases of potential interest in the form of a bar chart (figure 1c; revised).

Figure 1: Kinome profiling reveals increased activity of the CDK-Rb-E2F signaling pathway HPASMCs from IPAH patients. (a) Mean value of raw data for all individual samples, e.g. HPASMCs from healthy individuals (n=5) and IPAH patients (n=6), presented as a heat map of log-transformed fluorescence signals upon substrate serine/threonine phosphorylation that are used to perform an upstream kinase analysis. (b) Mean value of a two-group comparison for all samples showing the global changes in the serine/threonine phosphorylation pattern between both HPASMCs entities under basal media conditions. (c) Bar chart displaying the overall prediction of kinases with higher activity in IPAH-patient derived samples. Normalized kinase statistics is a mathematical based algorithm indicating the estimated relative kinase activity while the specificity score reflects the reliability and accuracy of the prediction.

Based on the above-mentioned analysis, we focused on CDKs as these data sets clearly demonstrates CDK6, CDK9, CDK4 and CDK2 with a highest prediction value in IPAH-derived HPASMCs (figure 1d; revised).

Figure 1. (d) Two-group comparison for the four selected CDKs only including the phosphorylation pattern of their particular peptide substrate sets on the peptide array.

For better clarity, we (i) changed the format of the two group comparison description, (ii) included the computational upstream kinase prediction in the revised version of the manuscript, and (iii) provided a phosphorylation profile for upstream kinase prediction of CDK2, CDK4, CDK6, and CDK9 (figure 1d; revised). In line, we have observed upregulation of two CDK-Rb-E2F downstream target genes namely CDK1 (figure 1f; revised) and Cyclin A2 (figure S2e; revised) in IPAH-patient derived HPASMCs, suggesting increased CDK activity IPAH-patients-derived HPASMC compared to healthy HPASMCs. As suggested by the reviewer, the resolution of the figures 1c and 1d has been increased for a better readability (figure 1c, 1d and S1; revised). Results referring to this question can be found in the revised version of the manuscript (page 6-7, line 118-149).

Figure 2.

-The Authors investigated the expression of CDK2, CDK4 and CDK6 in human tissues derived from healthy and IPAH patients. However, the Authors do not clearly explain the reason why they focused on these three CDKs.

R: We would like to thank the reviewer for this comment and the observation that several CDKs have been found to be active in our kinase screen. Since, altered cell cycle and an increased proliferative potential of IPAH-HPASMCs^{2,6} is the major driver of pulmonary vascular remodeling and CDK1, CDK2, CDK4 and CDK6 are considered as bona fide cell cycle regulators, we focused on them⁷. Importantly, among these cell cycle regulating CDKs, computational upstream kinase prediction displayed a highest prediction value of CDK2, CDK4 and CDK6 (but not of CDK1) in IPAH-derived HPASMCs (figure 1c; revised), instigating us to focus on CDK2, CDK4 and CDK6.

Figure 1. (c) Bar chart displaying the overall prediction of kinases with higher activity in IPAH-patient derived samples. Normalized kinase statistics is a mathematical based algorithm indicating the estimated relative kinase activity while the specificity score reflects the reliability and accuracy of the prediction.

This information has been added to the revised version of the manuscript on page 6, line 126-128.

Figure 3.

-The expression of CDK2, CDK4 and CDK6 on mRNA level (figure 3d) and protein level (figure 3e) is not consistent. For example, the brain and muscle tissues showed the highest mRNA levels of CDK4 when compared to the other organs, however CDK4 was undetectable on protein level in both brain and muscle. The Authors should comment on this aspect.

R: The discrepancy between the mRNA and its protein amount suggests regulation of CDKs at the post-transcriptional level. Post-transcriptional regulation of the specific miRs that influence expression of CDKs likely leads to an alternative mechanism(s) for regulation of their expression. For example, CDK2 expression can be post-transcriptionally suppressed by

miR-200c, whereas CDK4/ CDK6 expression can be post-transcriptionally suppressed by miR-6883-5p and miR-149*. These microRNAs directly bind to 3'UTR of CDKs and repress its translation^{8,9}. Thus, the regulation of CDK activity by specific miRNAs might be important both under physiological and pathological conditions. Thus, we concomitantly analyzed protein expression of CDK2/4/6/9 in all experimental settings (figure 1c, 3e-f, 6d, 8e, S5d, S9c).

Figure 4 and figure 5.

-In figure 4 and in figure 5a-c the Authors show the effect of the treatment with dinaciclib and palbociclib on cell proliferation, DNA synthesis etc, however it is not clear whether these experiments were performed by using HPASMC derived from healthy or IPAH patients. The Authors should clarify this aspect. Also, the Authors did not show any data relative to the measure of the IC50, for both the drugs, in their system.

R: In the submitted version of our manuscript, we used healthy HPASMCs to study the effects of dinaciclib and palbociclib on cell proliferation and DNA synthesis. As suggested by the reviewer, we performed additional experiments with HPASMCs isolated from lungs of IPAH-patients in order to study the efficacy of dinaciclib and palbociclib on cell proliferation and to determine the IC₅₀ value of the compounds (figure 3a-d; revised). Dinaciclib affects cell growth and survival of 47 diverse tumor cell lines with IC₅₀ values ranging from 6 to 17 nM¹⁰ which is in line with our findings for healthy HPASMCs (IC₅₀ of 12.3 nM; figure 3a; revised) and IPAH-HPASMCs (IC₅₀ of 1.75 nM; figure 3c; revised). Similar holds true for palbociclib which inhibited DNA synthesis in RB⁺ human colon and lung carcinoma as well as in leukemia cells with IC₅₀ values ranging from 40 to 170 nM¹¹. In healthy HPASMCs we determined an IC₅₀ value of 112 nM (figure 3b; revised) and in IPAH-HPASMCs an IC₅₀ of 29.2 nM (figure 3d; revised). This information has been added to the revised version of the manuscript as figure 3 and to the result section at page 8 (line 167-172).

Figure 3: Evaluation of IC_{50} concentrations of both CDK inhibitors, dinaciclib and palbociclib, and their effects on CDK-Rb-E2F signaling of human PSMCs of both origins. Cells were synchronized in basal media (BM) and then treated with the indicated concentrations of dinaciclib (upper panels) or palbociclib (lower panels) or DMSO as control in the presence of growth media (GM-2) for 24 h. IC_{50} graphs determining the half maximal inhibitory concentration of dinaciclib (upper panel) and palbociclib (lower panel) to block GM-2 induced proliferation of healthy HPASMCs (a, b) and of IPAH-HPASMCs (c, d) as measured by BrdU incorporation (relative absorbance [$A_{370nm} - A_{492nm}$]). IC_{50} values were calculated from measurements of two individual primary cell isolates (run twice in triplicates each) by the non-linear regression (curve fit) module with a variable slope using four parameters as provided by GraphPad Prism software.

-In addition, it is not clear why the Authors used as control the cells treated with basal media (BM) and not with GM-2 media as the cells treated with dinaciclib and palbociclib.

R: We apologize for the misleading labeling of previous figure 4 and 5 (figure 3 and 4; revised) suggesting that all experiments were done under basal medium conditions. As mentioned in the figure legend, all CDK inhibitor and their vehicle (DMSO) treatments were done in presence of GM-2. This has been clarified in the legend of the revised figures.

-In figure 5a and 5b the Authors show the effect of the treatment of the cells with dinaciclib and palbociclib on the cell cycle and cell apoptosis. In lines 195-197, the Authors state that there was no sign of cell death induction upon treatment of the cells with the drugs, since the fraction of cells in SubG1 phase did not increase (figure 5a-b). However, in lines 200-204 the Authors affirmed that dinaciclib triggered apoptosis-

mediated cell death (figure c). The Authors should explain this discrepancy. It is opinion of this reviewer that a deeper characterization of the apoptosis process, also under a molecular point of view (i.e. by investigation the potential involvement of caspases), would be necessary.

R: Taking a closer look at our data (figure 4e; revised), we want to point out that we only see a mild increase in the percentage of early apoptotic cells from 1.4% to 2.1% (AV⁺/PI⁻) and from 4.9% to 7.9% for late apoptotic cells (AV⁺/PI⁺) at a the highest concentration of dinaciclib but not of palbociclib (figure 4j, revised).

Figure 4. HPASMCs were synchronized in basal media (BM) and then treated with the indicated concentrations of dinaciclib (e) or palbociclib (j) or DMSO as control in the presence of growth media (GM-2) for 24 h. Combined staining of cells with AV and PI was detected by flow cytometry to reveal different stages of cell death due to dinaciclib (e) and palbociclib (j): early apoptosis (AV⁺PI⁻, yellow bars), late apoptosis (AV⁺PI⁺, orange bars), dead cells (AV⁻PI⁺, red bars), and living cells (AV⁻PI⁻, green bars).. Data from three independent experiments performed in triplicates are presented as mean±SEM depicting the percentage of the indicated populations. Statistical analysis was performed using one-way ANOVA with Dunnett’s post-hoc test to compare all bars versus DMSO control; *** p < 0.001.

As suggested by the reviewer, we performed Western blot analysis for Caspase-3 activation with an antibody specific for the cleaved form of Caspase-3. We could not see any expression of caspase-3 neither in healthy (figure 3e; revised) nor in IPAH-PASMCs at any given concentration (figure 3f; revised), suggesting no sign of cell death induction upon treatment of HPASMCs with both CDK inhibitors.

Figure 3. Representative Western blots for the detailed analyses of cellular signaling on protein level after 24 hours of treatment with dinaciclib, palbociclib, or staurosporine (SSP) in healthy HPASMCs (e) and IPAH-HPASMCs (f) with regard to CDK activation and downstream Rb-E2F pathway induction.

The statements with regard to cell death induction has been revised in the new version of the manuscript (page 8, line 182-183; page 9, line 211-219).

-In figure 5d the Authors show the effect of the treatment of healthy HPASMCs on the levels of phosphorylated and total Rb. Here, both dinaciclib and palbociclib reduced the levels of phosphorylated Rb (pRb), without affecting the levels of total Rb. By contrast, in the western blot shown in figure 5e, the levels of both pRb and total Rb were reduced in healthy HPASMCs treated with both dinaciclib and palbociclib. The Authors should explain this discrepancy. Also, concerning the levels of p21, the Authors do not comment on the reported differences between cells from healthy and pathological individuals or among cells treated with the different drugs. Finally, in figure 5d, the Authors show the detection of PCNA, which is not mentioned neither in the manuscript nor in the figure legend.

R: We would like to thank the reviewer for this comment. As already mentioned by reviewer 1, the sample heterogeneity of human cells isolated from donors and IPAH-lungs can contribute to the observed Rb downregulation. Assessment of Rb/P-Rb expression in healthy and IPAH-HPASMCs that are treated with different concentrations of dinaciclib and palbociclib suggest a concentration dependent decrease of Rb phosphorylation (figure 3; revised) which was strongly supported by downregulation of target genes CCNA2 and CDK1 (figure 3h and i; revised).

Figure 3. Representative Western blots for the detailed analyses of cellular signaling on protein level after 24 hours of treatment with dinaciclib, palbociclib, or staurosporine (SSP) in healthy HPASMCs (e) and IPAH-HPASMCs (f) with regard to CDK activation and downstream Rb-E2F pathway induction. (g) Western blot analysis for (P-)Rb and Cyclin D3 of healthy and IPAH-PASMCs treated with dinaciclib, palbociclib, or imatinib. CCNA2 (left) and CDK1 (right) mRNA expression (normalized to GAPDH as a house keeping gene) of healthy HPASMCs (h) and diseased IPAH-HPASMCs (i) upon 24 hours of inhibitor exposure. All data from two individual primary cell isolates (run twice in triplicates each) are presented as mean±SEM of the n-fold change ($2^{-\Delta\Delta C_t}$) compared with DMSO treated control samples (black bar). Statistical analysis was performed using one-way ANOVA with Newman-Keuls post-hoc test for multiple comparisons; *** p < 0.001. P-values for distinct conditions are only given for their comparison with DMSO-treated control cells (black bar), and the significance of the difference between the three conditions with various concentrations of each of the CDK inhibitors is represented as follows: § p < 0.05, §§ p < 0.01, §§§ p < 0.001.

We removed p21 as it is not a direct target gene of the CDK signaling pathway and included the PCNA data in the result and discussion section as well as in the figure legend.

-In the experiments shown in figure 5e the Authors introduced the use of imatinib, however they do not explain the rationale of using this drug and of comparing it to dinaciclib and palbociclib.

R: Imatinib is a well-known PDGFR/Abl/tyrosine kinase inhibitor which was the first kinase inhibitor that demonstrated anti-proliferative and anti-remodeling efficacy in experimental and clinical PAH^{12,13}. The rationale of using imatinib in this context is to compare the mode of action and underlying downstream molecular mechanisms of both CDK inhibitors. While we observed that both CDK inhibitors completely blocked Rb expression and activation in healthy- and IPAH-HPASMCs, imatinib did not appear to exert its known anti-proliferative effects by interfering with the Rb-E2F signaling pathway, suggesting the specific mode of action of CDK inhibitors. Due to extensive data obtained during revision process, we rearranged the figures and moved this data to another section (figure 3g; revised).

Figure 3. (g) Western blot analysis for (P-)Rb and Cyclin D3 of healthy and IPAH-PASMCs treated with dinaciclib, palbociclib, or imatinib.

Results referring to this question can be found in the revised version of the manuscript (page 8, line 175-177).

-It is opinion of this reviewer that the levels of phosphorylation of CDK2, CDK4, CDK6 and CDK9 and their total levels should be included in the western blot analysis. Indeed, since these CDKs are the targets of the used drugs, their characterization is crucial.

R: We appreciate this comment and addressed this concern accordingly. We performed additional Western blots to detect P-CDK2, P-CDK6, and P-CDK9 on protein level from (i) healthy and IPAH-HPASMCs (figure 1e; revised), (ii) in donor and IPAH-HPASMCs treated with different concentrations of dinaciclib and palbociclib (figure 3e, f; revised) and (iii) in lungs from palbociclib-treated MCT and Su/Hox animals (figure 6d and 8e; revised). Unfortunately, P-CDK4 antibodies are not commercially available.

Figure 1. (e) Western blot analysis for CDK2, CDK4, CDK6, and CDK9 activation and subsequent Rb-E2F downstream signaling in HPASMCs obtained from healthy donors and IPAH patients exposed to basal media.

Figure 3. Representative Western blots for the detailed analyses of cellular signaling on protein level after 24 hours of treatment with dinaciclib, palbociclib, or staurosporine (SSP) in healthy HPASMCs (**e**) and IPAH-HPASMCs (**f**) with regard to CDK activation and downstream Rb-E2F pathway induction.

Figure 6. (d) Western blot analysis for CDK2 and CDK6 activation and subsequent Rb-E2F downstream signaling in lung homogenates obtained representative samples from all three experimental groups e.g. healthy, MCT, MCT+Palbo.

Figure 8. (e) Western blot analysis for CDK2 and CDK6 activation and subsequent Rb-E2F downstream signaling in lung homogenates obtained representative samples from all three experimental groups e.g. Nox, Su/Hox, Su/Hox+Palbo.

These data suggest that there is increased phosphorylation of CDKs in IPAH-HPASMCS and in lungs from experimental models of PH that was decreased by palbociclib treatment.

Figure 7

-In figure 7i and 7j, the Authors show that the treatment of MCT-treated rats with palbociclib led to a reduction in the mRNA levels of CCN2A and CDK1. From these data the Authors concluded (lines 535-534) that "cellular proliferation was almost completely blocked in the animals treated with palbociclib, as evidenced by the reduced levels of two important transcriptional CDK-Rb-E2F downstream target genes, CCN2A and CDK1". It is opinion of this reviewer that the observed reduction in the mRNA levels of CCN2A and CDK1 is not enough to conclude that cell proliferation was blocked. Additional experiments, such as the immunohistochemical detection of proliferation markers like ki67 should be performed. Also, an evidence that palbociclib

actually affected the activity of the targeted CDK4 and CDK6 *in vivo* should be included. The levels of phosphorylated and total CDK4 and CDK6 should be determined (for example by immunohistochemistry).

R: We fully agree with the reviewer and performed PCNA staining of lung tissue from both animal models (MCT and Su/Hox animal model of PH) to assess the *in vivo* proliferation and measure the effects of palbociclib.

Figure 6. (c) Proliferation index as a quantitative measure of PCNA-positive cells (purple) per vessel. Representative images (using hematoxylin/eosin as counterstain) for all three study groups are included. Data are presented as mean±SEM and statistical analysis was performed using one-way ANOVA with Newman-Keuls post-hoc test for multiple comparisons; *** $p < 0.001$ for MCT+Palbo versus MCT; ### $p < 0.001$ for healthy versus MCT.

Figure 8. (d) Proliferation index as a quantitative parameter of PCNA-positive cells (purple) per vessel. Representative images (using hematoxylin/eosin as counterstain) for all three study groups are included. Data are presented as mean±SEM and statistical analysis was performed using one-way ANOVA with Newman-Keuls post-hoc test for multiple comparisons; * $p < 0.05$ for Su/Hox+Palbo versus Su/Hox.

In addition, we performed Western blots to detect (P-) CDK2 and (P-)CDK6, on protein level from the lungs of palbociclib-treated MCT and Su/Hox animals (figure 6d and 8e; revised).

Figure 6. (d) Western blot analysis for CDK2 and CDK6 activation and subsequent Rb-E2F downstream signaling in lung homogenates obtained representative samples from all three experimental groups e.g. healthy, MCT, MCT+Palbo.

Figure 8. (e) Western blot analysis for CDK2 and CDK6 activation and subsequent Rb-E2F downstream signaling in lung homogenates obtained representative samples from all three experimental groups e.g. Nox, Su/Hox, Su/Hox+Palbo.

These data suggest that palbociclib affects proliferation by inhibition of CDK6 phosphorylation leading to reduced levels of P-Rb and subsequent downregulation of the E2F transcriptional targets CCNA2, CDK1 (mRNA analysis; figure 6e, 8f; revised) and PCNA (proliferation index; figure 6c and d; 8d and e; revised).

Figure 6. (e) Analysis of CCNA2 (left) and CDK1 (right) mRNA expression normalized to GAPDH as reference gene in lung homogenates from all three experimental groups. Data are presented as mean±SEM of the n-fold change ($2^{-\Delta\Delta C_t}$) compared with a healthy control rat. Statistical analysis was performed using one-way ANOVA with Newman-Keuls post-hoc test for multiple comparisons; *** $p < 0.001$ for MCT+Palbo versus MCT; § $p < 0.01$ for MCT+Palbo versus healthy; #### $p < 0.001$ for healthy versus MCT.

Figure 8. (f) Analysis of CCNA2 (left) and CDK1 (right) mRNA expression normalized to GAPDH as reference gene in lung homogenates from all three experimental groups. Data are presented as mean±SEM of the n-fold change ($2^{-\Delta\Delta C_t}$) compared with a healthy control rat. Statistical analysis was performed using one-way ANOVA with Newman-Keuls post-hoc test for multiple comparisons; *** $p < 0.001$ for Su/Hox+Palbo versus Su/Hox; §§ $p < 0.01$, §§§ $p < 0.001$ for Su/Hox+Palbo versus Nox; #### $p < 0.001$ for Nox versus Su/Hox.

The data are now included in the revised version of the manuscript on page 11, line 266-274; page 12, line 295-301.

Minor points

-Line 219, please correct "transcriptional target genes of activated by E2F" with "transcriptional target genes of E2F".

R: This mistake is now corrected.

-In the Results section, lines 98-107, the Authors explained in a really detailed manner the experimental procedure followed for the acquisition of the data shown in figure 1c-1e. Maybe this part can be moved to the Material and Method section.

R: As suggested, we moved this to the supplement section (page 53-54, line 1167-1193).

-Figure 2, line 767: Please correct "(b-n)" with "(h-n)".

R: Done.

-Figure 5: Please make the text in figure 5a bigger.

R: We will make sure, that all figures and labels are at appropriate size.

References

- 1 Dabral, S. *et al.* Notch1 signalling regulates endothelial proliferation and apoptosis in pulmonary arterial hypertension. *The European respiratory journal* **48**, 1137-1149, doi:10.1183/13993003.00773-2015 (2016).
- 2 Savai, R. *et al.* Pro-proliferative and inflammatory signaling converge on FoxO1 transcription factor in pulmonary hypertension. *Nature medicine* **20**, 1289-1300, doi:10.1038/nm.3695 (2014).
- 3 Xu, J. *et al.* Glucagon-Like Peptide-1 Mediates the Protective Effect of the Dipeptidyl Peptidase IV Inhibitor on Renal Fibrosis via Reducing the Phenotypic Conversion of Renal Microvascular Cells in Monocrotaline-Treated Rats. *BioMed research international* **2018**, 1864107, doi:10.1155/2018/1864107 (2018).
- 4 Zheng, Z. *et al.* Chlorogenic acid suppresses monocrotaline-induced sinusoidal obstruction syndrome: The potential contribution of NFkappaB, Egr1, Nrf2, MAPKs and PI3K signals. *Environmental toxicology and pharmacology* **46**, 80-89, doi:10.1016/j.etap.2016.07.002 (2016).
- 5 Gomez-Arroyo, J. G. *et al.* The monocrotaline model of pulmonary hypertension in perspective. *American journal of physiology. Lung cellular and molecular physiology* **302**, L363-369, doi:10.1152/ajplung.00212.2011 (2012).
- 6 Li, X. *et al.* Notch3 signaling promotes the development of pulmonary arterial hypertension. *Nature medicine* **15**, 1289-1297, doi:10.1038/nm.2021 (2009).
- 7 Peyressatre, M., Prevel, C., Pellerano, M. & Morris, M. C. Targeting cyclin-dependent kinases in human cancers: from small molecules to Peptide inhibitors. *Cancers* **7**, 179-237, doi:10.3390/cancers7010179 (2015).
- 8 Lulla, A. R. *et al.* miR-6883 Family miRNAs Target CDK4/6 to Induce G1 Phase Cell-Cycle Arrest in Colon Cancer Cells. *Cancer research* **77**, 6902-6913, doi:10.1158/0008-5472.CAN-17-1767 (2017).
- 9 Wang, X. *et al.* miR-200c Targets CDK2 and Suppresses Tumorigenesis in Renal Cell Carcinoma. *Molecular cancer research : MCR* **13**, 1567-1577, doi:10.1158/1541-7786.MCR-15-0128 (2015).
- 10 Parry, D. *et al.* Dinaciclib (SCH 727965), a novel and potent cyclin-dependent kinase inhibitor. *Molecular cancer therapeutics* **9**, 2344-2353, doi:10.1158/1535-7163.MCT-10-0324 (2010).
- 11 Fry, D. W. *et al.* Specific inhibition of cyclin-dependent kinase 4/6 by PD 0332991 and associated antitumor activity in human tumor xenografts. *Molecular cancer therapeutics* **3**, 1427-1438 (2004).
- 12 Hooper, M. M. *et al.* Imatinib mesylate as add-on therapy for pulmonary arterial hypertension: results of the randomized IMPRES study. *Circulation* **127**, 1128-1138, doi:10.1161/CIRCULATIONAHA.112.000765 (2013).
- 13 Schermuly, R. T. *et al.* Reversal of experimental pulmonary hypertension by PDGF inhibition. *The Journal of clinical investigation* **115**, 2811-2821, doi:10.1172/JCI24838 (2005).

Reviewers' comments:

Reviewer #1 (Remarks to the Author):

The authors have provided substantial new data in response to the original comments of the reviewers. These data clarify the majority of the outstanding questions, but raise a few additional questions.

1. The new data in Fig. 2 is much appreciated and is very clear with respect to expression of the CDKs in the vasculature of human iPAH. The interesting finding that expression of CDK2 and CDK4 seems to be highest in the adventitial cells while CDK 6 appears to be highest in the endothelium. Expression of these CDKs doesn't appear to be much greater in iPAH vs controls. Since you perform the kinase profiling studies with PSMCs, this suggests that either increased expression of the CDKs isn't important and/or PSMC CDK activity is activated by mediators from other cell types. This apparent disconnect should be acknowledged.
2. The response to Q5 is appreciated. The original question also asked for a rationale pertaining to the toxicity of one of the drugs as they share the same molecular pathway as outlined in Fig 9. Does it have anything to do with targeting CDK 1/5 as well?
3. The decline in neutrophils with CDK inhibitors should be acknowledged as a potential limitation for longterm clinical use.
4. A quick search shows that P-CDK4 antibodies are available from international companies.

Reviewer #2 (Remarks to the Author):

The Authors addressed all the concerns raised in my original review and properly revised the manuscript.

Point-by-point response to the comments of reviewer #1

Q1. The new data in Fig. 2 is much appreciated and is very clear with respect to expression of the CDKs in the vasculature of human iPAH. The interesting finding that expression of CDK2 and CDK4 seems to be highest in the adventitial cells while CDK 6 appears to be highest in the endothelium. Expression of these CDKs doesn't appear to be much greater in iPAH vs controls. Since you perform the kinase profiling studies with PASMCs, this suggests that either increased expression of the CDKs isn't important and/or PASMC CDK activity is activated by mediators from other cell types. This apparent disconnect should be acknowledged.

R1: We appreciate the thorough evaluation of our IHC stainings of human specimens. We agree with the reviewer that the intensity of CDK2 and CDK4 immunoreactivity seems to be highest in the adventitial cells while CDK 6 appears to be highest in the endothelium and immunoreactivity of CDKs is not strikingly altered in IPAH. However, interestingly

- (i) direct targets of CDKs such as p-Rb and PCNA immunoreactivity is strongly increased in all vascular cells of IPAH.
- (ii) peptide-based kinase activity profiling reveals increased activity of the CDK-Rb-E2F signaling pathway in PASMCs from IPAH patients.
- (iii) prominent expression of CDKs was observed in human PASMCs and a significant change in the pattern of phosphorylation of CDKs was noted when comparing the healthy PASMCs and IPAH-PASMCs.
- (iv) dinaciclib and palbociclib, two CDK activity inhibitors employed in this study, decreased the proliferation of both PASMCs and PAECs.

Considering all the above facts, we agree with the reviewer that increased pulmonary vascular CDK activity is a hallmark of IPAH patients, which is driven by secreted growth factors from PASMCs or other vascular cells. We believe that increased CDK expression may occur at an early time point, and as suggested by reviewer acknowledge this in the revised manuscript on page 15 in line 377-384: *“Although we focused primarily on the differential CDK activity in HPASMCs due to our initial data obtained by peptide-based kinase activity profiling, we could detect expression of CDK2, CDK4 and CDK6 also in other pulmonary vascular cell types (i.e. endothelial and immune cells, adventitial fibroblasts) in the vasculature of both donor and IPAH patients. However, increased immunoreactivity of the CDK targets P-Rb and PCNA was observed mainly in IPAH remodeled vasculature, suggesting increased CDK activity in IPAH is a prominent phenomenon driven by elevated secretion of growth factors from PASMCs or other vascular cells”.*

Q2. The response to Q5 is appreciated. The original question also asked for a rationale pertaining to the toxicity of one of the drugs as they share the same molecular pathway as outlined in Fig 9. Does it have anything to do with targeting CDK 1/5 as well?

R2: We thank the reviewer for this question, which we would like to answer in more detail with respect to the molecular mode of action for both drugs. Dinaciclib is indeed targeting several other CDKs with comparable affinities: for CDK2 this is reflected by low IC₅₀ values of 1 nM (CDK2 and CDK5) and 3 nM (CDK1) and 4 nM (CDK9) as determined in cell-free assays. CDK2 and CDK1 are required for S-, G2-, and M-phase progression ¹. Furthermore, dinaciclib is able to inhibit the unfolded protein response (UPR) via CDK1 and CDK5 which links these components of the cell-cycle regulatory apparatus (e.g. CDK1/5) with the cytoprotective mechanisms of the UPR ². In our manuscript, we determined the effects of both compounds, dinaciclib and palbociclib, on CDK1 mRNA expression by real-time PCR analyses as a consequence of any disruption of the CDK-Rb-E2F signaling pathway. Here, a remarkable reduction of CDK1 levels was observed (figure 1f 3h, 3i, 6e, 8f, S5e; revised). The function of CDK9 in regulating gene transcription elongation and mRNA maturation, as well as other physiologic processes was already addressed in the former revision of the manuscript (reviewed in ³⁻⁵). We investigated the mRNA expression of CDK9 in human specimen (figure S2d, S2l; revised) and experimental models of P(A)H (figure S3d, S3h, S3l; revised). Only in the animal disease models, a significant change under disease conditions could be observed. This indicated that during the development of P(A)H in those animal models CDK9 plays an important role and that any interference with its activity might be crucial. In contrast, in Western blot analyses for the phosphorylation of CDK9 (using a P-CDK9 specific antibody) we did not observe any changes in its activation status upon dinaciclib or palbociclib treatment of isolated primary human pulmonary artery smooth muscle (figure 3e, 3f; revised) and endothelial cells (figure S5d; revised). These facts might explain the high efficacy of dinaciclib on cellular proliferation *in vitro* (figure 4a, 4f, S5a; revised) but also its fatal side-effects *in vivo* in terms of increased mortality. Palbociclib targets CDK4 and CDK6 with high specificity and in low nanomolar concentrations of 11 nM and 16 nM in cell-free assays, respectively. No activity was reported for any other CDKs. Therefore, the effects of palbociclib reported in this manuscript are a direct and exclusive consequence of a targeted cell cycle inhibition via CDK4 and CDK6. In figure 9, we summarized the signaling of pulmonary vascular cells and the involvement of CDKs only in the cell cycle in a very schematic and simplified manner. We strongly focused on the investigation of the CDK-Rb-E2F signaling pathway in human pulmonary vascular cells and the possible differences in the phenotype in regard to proliferation and survival *in vitro* and *in*

vivo (figure 5, 6, 7, 8; revised). We now added the following statement to the revised manuscript on page 17 in line 427-431: *“Besides this, dinaciclib has a broader spectrum of activities as compared to palbociclib, which encompasses cell cycle inhibition, interference with protein synthesis (via CDK9³⁻⁵) and the regulation of the unfolded protein response (via CDK1 and CDK5²). This might also explain the highly toxic profile of dinaciclib in vivo especially in combination with MCT.”*

Q3. The decline in neutrophils with CDK inhibitors should be acknowledged as a potential limitation for long-term clinical use.

R3: Myelosuppressive effects (especially neutropenia) are common side-effects for palbociclib treatment similar to many chemotherapeutics that aim to inhibit cellular proliferation as. For the treatment of women with HR⁺, HER2⁻ metastatic breast cancer, Ibrance capsules (containing palbociclib) are taken orally with food in combination with an aromatase inhibitor or fulvestrant. The recommended starting dose is 125 mg once daily taken with food for 21 days followed by 7 days off treatment. This dosing interruption and/or dose reductions (down to 75 mg) are recommended based on individual safety and tolerability. Palbociclib is considered to be a reversible CDK inhibitor and studies showed that the neutrophil count recovers during the intervals between the treatment cycles⁶. Thus, monitoring the neutrophil blood count would be the ideal parameter to observe unwanted toxic side-effects and to decide about dose reduction or interruption. Until now, it is unclear which time period is thought to be needed to lead to a stop (or reversal) of the pulmonary vascular remodeling process in PAH. A proper readout to monitor palbociclib efficacy is right-heart catheterization by measure PVR. In the IMPRES trial, PAH patients showed an impressive improvement in the pulmonary vascular resistance and six minute walking distance indicating the beneficial effects already after 24 weeks of Imatinib treatment⁷. In oncologic malignancies, kinase inhibitors like Imatinib in general are administered as long as there is no evidence of progression of the disease or of unacceptable toxicity. These data don't contradict a long-term use of this class of compounds per se. But considering the occurrence of neutropenia and the given regimen in form of repetitive cycles, an uncritical long-term clinical use of palbociclib is very likely not possible. To circumvent any of the known adverse events, a dose reduction carried out by a combined therapy with approved PAH drugs with conventional vasodilatory effects might be the first option. The possible limitation of palbociclib in long-term clinical use is now acknowledged in the newly revised manuscript on page 19 in line 474-477: *“Neutropenia, known to be an undesirable side effect of palbociclib and potentially limiting a long-term clinical use, could be circumvented by dose*

adaptation (as previously mentioned), combined therapy with other approved PAH drugs and/or by local administration (i.e. inhalation).”

Q4. A quick search shows that P-CDK4 antibodies are available from international companies.

R4: We were intensively searching for adequate antibodies for all CDKs in this study and had difficulties to find antibodies specifically detecting the phosphorylated forms for Western blot analysis. Due to high sequence homology it is almost impossible to generate antibodies that clearly distinguish the different isoforms. We tested several of them using several techniques for immunohistological stainings of human and rat lung specimen. Unfortunately, we could not find any phospho-specific antibodies (i.e. for P-CDK2 and P-CDK6) which resulted in reliable and trustworthy data. In particular for P-CDK4, no antibody was available at the time of our analysis and until today, only one antibody from *Mybiosource* (#MBS9126724) or *LSBio* (LS-C411605) for Western blot analysis of human P-CDK4 with an unknown performance (that means quality and specificity) for other applications and species can be obtained. For this reason, we focused on Western blot analyses of the direct downstream targets P-Rb and PCNA as appropriate readouts to evaluate the inhibitory effects of dinaciclib and palbociclib on CDK activity.

References

- 1 Roskoski, R., Jr. Cyclin-dependent protein serine/threonine kinase inhibitors as anticancer drugs. *Pharmacological research*, doi:10.1016/j.phrs.2018.11.035 (2018).
- 2 Nguyen, T. K. & Grant, S. Dinaciclib (SCH727965) inhibits the unfolded protein response through a CDK1- and 5-dependent mechanism. *Molecular cancer therapeutics* **13**, 662-674, doi:10.1158/1535-7163.MCT-13-0714 (2014).
- 3 Boffo, S., Damato, A., Alfano, L. & Giordano, A. CDK9 inhibitors in acute myeloid leukemia. *Journal of experimental & clinical cancer research : CR* **37**, 36, doi:10.1186/s13046-018-0704-8 (2018).
- 4 De Falco, G. & Giordano, A. CDK9: from basal transcription to cancer and AIDS. *Cancer biology & therapy* **1**, 342-347 (2002).
- 5 Bres, V., Yoh, S. M. & Jones, K. A. The multi-tasking P-TEFb complex. *Current opinion in cell biology* **20**, 334-340, doi:10.1016/j.ceb.2008.04.008 (2008).
- 6 Verma, S. *et al.* Palbociclib in Combination With Fulvestrant in Women With Hormone Receptor-Positive/HER2-Negative Advanced Metastatic Breast Cancer: Detailed Safety Analysis From a Multicenter, Randomized, Placebo-Controlled, Phase III Study (PALOMA-3). *The oncologist* **21**, 1165-1175, doi:10.1634/theoncologist.2016-0097 (2016).
- 7 Hoeper, M. M. *et al.* Imatinib mesylate as add-on therapy for pulmonary arterial hypertension: results of the randomized IMPRES study. *Circulation* **127**, 1128-1138, doi:10.1161/CIRCULATIONAHA.112.000765 (2013).

REVIEWERS' COMMENTS:

Reviewer #1 (Remarks to the Author):

The authors have provided a nice response to comments raised in prior review. The explanations are concise and the text added is explanatory.